# Enhancing symbolic image classification through Gaussian copulas and optimized distinguishing points

Sri Winarni[1]*, Sapto Wahyu Indratno[2], Mohd Shahizan Othman[3],
Siti Zaiton Mohd Hashim[3], Mohd Murtadha Mohamad[3], Apri Junaidi[3], Ebenezer Bonyah[4,5],
Anindya Apriliyanti Pravitasari[1], Triyani Hendrawati[1], Irlandia Ginanjar[1]

1 Department of Statistics, Universitas Padjadjaran, Sumedang, Indonesia, 2 Statistics Research Group, Institut Teknologi Bandung, Bandung, Indonesia, 3 Faculty of Computing, Universiti Teknologi Malaysia, Johor Bahru, Malaysia, 4 Department of Mathematics Education, Akenten Appiah-Menka University of Skills Training and Entrepreneurial Development, Kumasi, Ghana, 5 Azerbaijan University of Architecture and Construction, Baku, Azerbaijan

* sri.winarni@unpad.ac.id

## Abstract

In this paper a new method is being proposed which makes use of symbolic data manifested by empirical cumulative distribution functions (ECDF) and distribution functions based on sets of ECDF values, referred to as the distribution function of distribution values (DFDV) of image features. This differs with conventional image classification studies, which mostly rely on pixel intensity values as features. Such symbolic representations provide a more general characterization of the pixel intensities patterns across image areas. The main novelty associated with the given research is the creation of the DFDV based on the best possible selection of the distinguishing points in order to capture fundamental changes in intensity distributions within image classes. Compared to previous studies which have used pre-determined distinguishing points, the proposed study proposes a clustering-based approach to find distinguishing points that maximise class separability. Experimental evaluation on the MNIST handwritten digits data set demonstrates the effectiveness of the proposed method, achieving an average classification accuracy of 68.27% and a highest accuracy of 95.33%. These results indicate that integrating clustering-based symbolic features extraction with copula-based modelling provides a competitive and promising for image classification tasks.

## Introduction

Image classification is the branch of artificial intelligence (AI) as a fundamental type of task used in different industries, such as healthcare, security, or automation. Due to the rapid development of technology, there is a considerable rise in the need of effective and accurate classification methods to contribute to the activities of object identification, numerical image analysis and plant disease identification [1–6]. The

**Data availability statement:** The MNIST dataset is publicly available from its original source at http://yann.lecun.com/exdb/mnist/. All author-generated MATLAB source code necessary to reproduce the findings of this study is publicly available at Zenodo (https://doi.org/10.5281/zenodo.18742616).

**Funding:** This research received funding from the Internal Matching Funds Research Grant (IMF) at Universitas Padjadjaran, Indonesia, for the project "Model Classification of Rice Plant Diseases Based on Deep Learning and Gaussian Copula to Support Sustainable Precision Agriculture," under contract number 4356/UN6.D/PT.00/2025.

**Competing interests:** The authors disclose no potential conflicts of interest regarding the research, authorship, or publishing of this work.

recognition of handwritten digits is of exceptional importance in this collection of applications because it is generally used in automation procedures like cheque scanning, document reading, and data entering. Miscalculations in numbers detection can lead to major impact on financial security, lower productivity, and compromised quality of an activity. It has found relevance in cybersecurity too, most notably verification of user inputs in captchas, authenticity of handwritings to verify handwritten signatures, and fraudulent numerical codes [7,8]. As a result, the modern studies are aimed at creation of new methods to enhance the correctness, stability, and data reliability of the image classification systems. The increase in the implementation of reliable and accurate classification techniques remains a major research field.

There are a number of remedies which have been suggested over the past few years to enhance performance of image classification. The use of deep learning has created enormous excitement due to the capacity to learn complicated and even multidimensional features on image data [9–11]. Typically, such methods have significant processing requirements and need huge numbers of labelled data; also, their core processes are generally hard to interpret [12–15]. Probabilistic, on the other hand, such as the Gaussian copula build a more explanatory framework of modelling since there is an explicit representation of the dependencies between features in a statistical basis. Although this method is rather rare in image classification, it has a considerable potential, especially taking into account the importance of interpretability and statistical structure rather than simplicity of the model.

The Gaussian copula is a performance that links the marginals distribution of a number of random components to create their joint distribution. The dependencies between several variables are determined by a multivariate normal distribution [16,17]. Gaussian copulas would be advantageous to image classification in explaining the associations between the features of the images and the labels. The advantage of this is specifically useful in simulating complex image data like handwritten digits which show variations in style and thickness. The given paper explores the use of symbolic information, in the form of distribution functions, symbolising image features treated as random variables within the Gaussian copula framework. The methodology differs with the usual analysis where pixel intensity information is used as features.

Symbolic data are data that carry information beyond numerical values and that may be categories, functions or distributions with intrinsic meanings [18,19]. In this work the symbolic data is used as an aspect of visual feature by treating pixel intensities breakdown as distributions of picture characterised as stochastic variables. Such an approach allows a more comprehensive modelling of statistical data compared to the use of raw pixel value or simple feature vectors. It has several advantages: (1) summarising of random variables behaviour in a convenient form of distribution; (2) reinforcement of stability in feature representation; (3) compacting data structure, which facilitates statistical analysis. Thus, the method can serve as a relevant alternative to the statistical modelling of image data.

The prerequisite of previous studies involving use of copulas in image classification systems is characterised by a heavy reliance on pixel intensity values and their variation attributes [20–23]. A symbolic data technique using distribution functions in

particular the empirical cumulative distribution function (ECDF) and the distribution function of distribution values (DFDV) as an image feature proposed itself in a recent study [24]. In that study, DFDV was developed utilising a predetermined set of distinguishing points to encapsulate essential elements of the ECDF. Nonetheless, employing fixed distinguishing points may constrain the capacity to identify the most pertinent variances among image categories. This study enhances [25] which previously achieved an accuracy of 62.66%, by introducing an innovative strategy for identifying distinguishing points using clustering. This adaptive technique seeks to discern image features that more efficiently distinguish classes by identifying salient patterns in the data. The dynamic selection of differentiating features is anticipated to enhance classification accuracy by more accurately representing the statistical structure of the images.

To evaluate the proposed framework, experiments are conducted using the MNIST handwritten digit dataset, which remains a widely adopted benchmark for image classification research. Recent MNIST studies predominantly focus on optimizing predictive performance through convolutional neural networks, recurrent models, and hybrid deep learning architectures [26–30], typically relying on pixel-level representations and increasingly complex neural structures. In contrast, this study does not aim to introduce a new neural architecture, but rather to demonstrate a fundamentally different statistical modeling perspective. MNIST was chosen as it is a standard dataset for image classification, facilitating performance comparison with previous studies. Its variations in handwriting and pixel intensity reflect real-world challenges, allowing methods that perform well to be adapted for applications such as OCR and symbol recognition.

The main novelty of this work lies in integrating optimized symbolic feature construction based on ECDF with Gaussian copula modeling within a unified classification framework. Specifically, the proposed approach constructs DFDV using clustering-based optimized distinguishing points and captures dependence structures among symbolic features through copula modeling. This integrated framework provides a statistically grounded and interpretable alternative to conventional pixel-driven methods.

The study aims to (1) develop a copula model utilising distribution functions as random variables; (2) emphasise the discriminative capacity of the ECDF and the DFDV as salient image features; and (3) improve class separability via a clustering-based methodology for identifying optimal distinguishing points. This research's primary contributions are: (1) the creation of a Gaussian copula-based framework tailored for image classification tasks; (2) the incorporation of pixel intensity distributions as symbolic features to enhance classification performance; and (3) the progression of probabilistic methods in symbolic data modelling. The subsequent sections of this work are structured as follows. Section 1 introduces the history, rationale, objectives, and importance of incorporating Gaussian copula modelling in image classification. Section 2 delineates the methodology, encompassing the theoretical framework of Gaussian copula, the formulation of symbolic data via ECDF and DFDV, and the suggested clustering-based approach for pinpointing discriminative spots. Section 3 details the studies performed with the MNIST dataset of handwritten digits and delineates the classification results achieved with the proposed strategy. Section 4 brings forward the results discussion and analysis, which includes interpretation of the performance and comparison with other studies. The last section, section 5 provides an overall conclusion on the study by highlighting the important findings of the study in addition to suggesting areas of future research.

## Methods

In this section, the methodology of this research is outlined. The chapter starts by performing a clear definition of the Gaussian copula, which is expected to be used to define dependencies among random variables. The following subsections explain how symbolic data characteristics were generated via ECDF and DFDV, determining query points of difference, methodology of parameter estimation, and classification procedure as to the proposed copula-based model.

### Overview of gaussian copula

Gaussian copula is a mathematical construction, which defines the dependence relationship between the random variables, making the independent of their marginal distributional forms by simply linear transforming of multivariate normal

distributions. Copula provides a strong model to express the relation between multivariate random variables because it creates clear distinction between dependence structure and distribution of individual variables. The methodology follows the copula theory whereby a multivariate distribution is conceived as an integration of the marginal distributions of the same with a copula. In the multivariate normal case, the Gaussian copula easily approximates the linear correlations and creates scope to allow the marginal distributions to differ with normal distribution [31].

Consider $p$ random variables $(X_1, X_2, \ldots, X_p)$, each with a marginal CDF $F_i(x_i)$. Gaussian copula transforms these variables to uniform random variables $U_i = F_i(X_i)$ with respect to the range [0, 1]. The transformed variables retain the dependency patterns imbedded in the copula. We can mathematically articulate the joint CDF of random variables $X_i$ with marginal distributions $F_i$, as follows:

$$F(x_1, x_2, \ldots, x_p) = C(u_1, u_2, \ldots, u_p) \tag{1}$$

The copula function with $C$ is the function that generalizes the dependence between the referred to variables. The theory of copula has its theoretical footing in the Sklars theorem. It postulates that any distribution of multivariate can be presented as a copula which is in observance jointed with the marginal distributions [32]. In particular:

$$F(x_1, x_2, \ldots, x_p) = C(F_1(x_1), F_2(x_2), \ldots, F_p(x_p)) \tag{2}$$

where $u_i = F_i(x_i)$. This separation allows us to model the dependence structure, as shown in $C$, independently of the marginals $F_i(X_i)$. For completely continuous distributions, the associated density function is defined as:

$$f(x_1, x_2, \ldots, x_p) = c(u_1, u_2, \ldots, u_p) \prod_{i=1}^{p} f_i(x_i) \tag{3}$$

with $f_i(x_i) = \frac{dF_i(x_i)}{dx_i}$ is the marginal density, and $c(u_1, u_2, \ldots, u_p) = \frac{\partial^p C(u_1, u_2, \ldots, u_p)}{\partial u_1 \partial u_2 \ldots \partial u_p}$ is the copula density. The multivariate normal distribution forms the Gaussian copula. Let $(Z_1, Z_2, \ldots, Z_p)$ represent a vector of standard normal random variables characterised by a covariance matrix $\Lambda$. The Gaussian copula exhibits the following characteristics:

$$C_{\Lambda}(u_1, u_2, \ldots, u_p) = \Phi_{\Lambda}(Z_1, Z_2, \ldots, Z_p) \tag{4}$$

Here, $\Phi_{\Lambda}$ is the cumulative distribution function (CDF) of the multivariate normal distribution defined by the covariance matrix $\Lambda$, and $Z_i = \Phi^{(-1)}(u_i)$, where $\Phi^{(-1)}$ represents the inverse of the standard normal CDF, with $u_i \in [0, 1]$ for $i = 1, 2 \ldots, p$ signifying the marginal uniform variables [33].

The dependency structure in the Gaussian copula is defined by the covariance matrix $\Lambda$ of a multivariate normal distribution. This structure allows it to capture and represent linear relationships among the variables. At the same time, the marginal distributions of the variables are flexible and do not need to follow a normal distribution, enabling diverse applications with customized marginal properties. However, they are limited in capturing tail dependencies, as the Gaussian structure inherently assumes symmetric dependency patterns. This restriction is in contrast to the $t$-copula, and other copulas that have the ability of capturing greater tail dependencies. The Gaussian copula provides a density function $C(u_1, u_2, \ldots, u_p)$ for dependency modeling:

$$c(u_1, u_2, \ldots, u_p) = \frac{|\Lambda|^{-\frac{1}{2}}}{(2\pi)^{\frac{p}{2}}} exp\left(-\frac{1}{2} z^T \Lambda^{-1} z\right) \prod_{i=1}^{p} \frac{1}{\phi(\Phi^{-1}(u_i))} \tag{5}$$

In which $u_i$, $i = 1, 2, \ldots, p$, are uniform random variables that are related to the original data through the marginal cumulative distributions, $\mathbf{z} = \left(\Phi^{-1}(u_1), \Phi^{-1}(u_2), \ldots, \Phi^{-1}(u_p)\right)^T$ is the vector of inverse standard normal transformations, and $\Lambda$ is the covariance matrix with ones along the diagonal. $\phi$ is the standard normal density function, whereas $\Phi$ signifies its cumulative distribution function (CDF). In this formulation there is a normalisation term to cover fixed marginal distributions [20]. The joint density is altered using the equation $\prod_{i=1}^{p} \phi\left(\Phi^{-1}(u_i)\right)$ to make the marginal distributions fit a uniform distribution on the range [0,1]. Note that $\phi\left(\Phi^{-1}(u_i)\right) = \phi(z_i) = \frac{1}{\sqrt{2\pi}}\exp\left(-\frac{1}{2}z_i^2\right)$; hence,

$$\prod_{i=1}^{p} \phi\left(\Phi^{-1}(u_i)\right) = \frac{1}{(2\pi)^{\frac{p}{2}}} exp\left(-\frac{1}{2}\sum_{i=1}^{p} z_i^2\right) = \frac{1}{(2\pi)^{\frac{p}{2}}} exp\left(-\frac{1}{2}\mathbf{z}^\top \mathbf{z}\right)$$

(6)

Then by putting (6) in density function (5) hence, we get. Focusing on dependency structure of the Gaussian copula, the term $\frac{1}{2}\mathbf{z}^\top \mathbf{z}$ may be eliminated to arrive to:

$$c(u_1, u_2, \ldots, u_p) = |\Lambda|^{-\frac{1}{2}} exp\left(\frac{1}{2}\mathbf{z}^T\left(I - \Lambda^{-1}\right)\mathbf{z}\right)$$

(7)

$$\propto |\Lambda|^{-\frac{1}{2}} exp\left(-\frac{1}{2}\mathbf{z}^T\Lambda^{-1}\mathbf{z}\right)$$

(8)

The proportionality denotes the missing of the marginal density elements needed to uphold the regularity of the marginals. This is simplification, which points out to the ability of the copula to describe dependencies regardless of the specific marginal distributions. The joint density function of the Gaussian copula is expressed as follows:

$$f(x_1, x_2, \ldots, x_p) = |\Lambda|^{-\frac{1}{2}} exp\left(\frac{1}{2}\mathbf{z}^T\left(I - \Lambda^{-1}\right)\mathbf{z}\right)\prod_{i=1}^{p} f_i(x_i)$$

(9)

## Parameter estimation and marginal modeling

Gaussian copula models are often estimated using the maximum likelihood (ML) approach. In this strategy, both the marginal and copula parameters are combinedly computed. It means optimising the likelihood function of all parameters including the marginal distributions ($\Theta$) and the covariance matrix ($\Lambda$). But this method does not scale well as the space of the random variables in the model becomes dimensional, that is, growing exponentially with the number of random variables. The process of optimising the whole log likelihood requires large calculations especially in the cases of high dimensional data sets. The maximum likelihood two-stage inference of marginals (IFM) applied [34], usually deals with this difficulty. This method saves a lot of computational burden and at the same time estimates parameters correctly. The whole likelihood function for a Gaussian copula model can be articulated as follows:

$$L(\Theta, \Lambda | x_1, x_2, \ldots, x_p) = \prod_{j=1}^{N} |\Lambda|^{-\frac{1}{2}} exp\left\{\frac{1}{2}\mathbf{z}_j^T\left(I - \Lambda^{-1}\right)\mathbf{z}_j\right\}\prod_{j=1}^{N}\prod_{i=1}^{p} f_i(x_{ji})$$

(10)

In which $|\Lambda|$ the determinant of the covariance matrix is denoted as where, $z_j = \left(\Phi^{-1}\left(u_{j,1}\right), \ldots, \Phi^{-1}\left(u_{j,p}\right)\right)$ with $u_{j,i} = F_i\left(x_{j,i}\right)$, and $f_i\left(x_{ji}\right)$ is the marginal density of the i-th variable of the j-th data [35]. The associated log-likelihood function can be articulated as:

$$\ell\left(\Theta, \Lambda | x_1, x_2, \ldots, x_p\right) = \sum_{j=1}^{N} log|\Lambda|^{-\frac{1}{2}} exp\left\{\frac{1}{2}z_j^T\left(I - \Lambda^{-1}\right)z_j\right\} + \sum_{j=1}^{N}\sum_{i=1}^{p} log\, f_i\left(x_{j,i}\right)$$

(11)

The IFM approach divides the estimating procedure into two phases: The marginal parameters ($\Theta$) are initially computed for each variable individually. Subsequently, the copula parameters, particularly the covariance matrix ($\Lambda$), are estimated while the marginal parameters remain unchanged. We first derive the marginal parameters by optimising the likelihood function for the marginal distributions. The marginal parameters $\hat{\theta}$ are obtained by the marginal likelihood $\ell_f$.

$$\hat{\theta} = Argmax\, \ell_f$$

(12)

where $\ell_f = \sum_{j=1}^{N}\sum_{i=1}^{p} log f_i\left(x_{j,i}\right)$. The computed marginal parameters serve as inputs for the copula likelihood function, facilitating the estimation of the copula parameters. The estimate of copula parameters is accomplished by optimising the likelihood function $\ell_c$.

$$\hat{\Lambda} = Argmax\, \ell_c$$

(13)

$\ell_c = \sum_{j=1}^{N} log|\Lambda|^{-\frac{1}{2}} exp\left\{\frac{1}{2}z_j^T\left(I - \Lambda^{-1}\right)z_j\right\}$. The joint log-likelihood function is expressed as follows:

$$\ell\left(\Theta, \Lambda | x_1, x_2, \ldots, x_p\right) = \ell_c + \ell_f$$

(14)

where $\Theta = \left[\theta_1, \ldots, \theta_p\right]$ signifies the marginal parameters and $\Lambda$ represents the covariance matrix that parameterises the copula. The optimisation process commences with the estimation of marginal parameters via $\frac{\partial \ell_f}{\partial \Theta} = 0$. Upon determining $\hat{\Theta}$ these estimations are incorporated into the copula likelihood function $\ell_c$. The copula parameters are subsequently determined by resolving $\frac{\partial \ell_c}{\partial \Lambda} = 0$. Collectively, these fulfil the system's requirements:

$$\left(\frac{\partial}{\partial \Theta}\ell_f, \frac{\partial}{\partial \Lambda}\ell_c\right) = (0, 0)$$

(15)

In the Gaussian copula framework, marginal distributions may be estimated by either parametric or non-parametric methods, contingent upon the characteristics of the data. Kernel density estimation (KDE) is a prevalent non-parametric technique that estimates the probability density function of a random variable without presupposing any particular parametric structure. KDE operates by interpolating observable data points to generate a continuous representation of the underlying distribution. The KDE for a given set of data points is articulated as:

$$\hat{f}(y) = \frac{1}{N}\sum_{j=1}^{N} K\left(\frac{y - y_j}{h}\right)$$

(16)

Let $N$ signify the total number of data points, $K$ indicate the kernel function, $y_j$ represent the $j$-th observed data point, and $h$ imply the bandwidth parameter [36]. The choice of bandwidth influences the continuity of the density function estimation. Augmenting bandwidth yields more fluid and generalised estimates, whereas reducing bandwidth produces sharper and more reactive estimates that discern small intricacies in the data. KDE is advantageous when parametric

assumptions regarding marginal distributions are inadequate or when the data exhibits intricate structures that parametric distributions cannot adequately represent, due to its inherent flexibility.

## Symbolic feature construction with ECDF and DFDV

The ECDF serves as a statistical tool to characterise the distribution of pixel intensities in an image. For a certain image $j$, let $X_j$ represent a random variable corresponding to the pixel values, where $x = 0, 1, 2, \ldots, 255$. The ECDF, denoted as $\mathcal{F}_j(x)$, is computed using the following formula:

$$\mathcal{F}_j(x) = \frac{1}{n} \sum_{i=1}^{n} \mathbf{1} \left\{ x_{ji} \leq x \right\}$$

(17)

In this equation, $n$ denotes the total pixel count in the image, $x_{ji}$ indicates the intensity of the $i$-th pixel in the $j$-th image, and $\mathbf{1}\left(x_{ji} \leq x\right)$ is an indicator function that equals 1 when the pixel intensity $x_{ji}$ is less than or equal to $x$, and 0 otherwise [37].

The ECDF function $\mathcal{F}_j(x)$ represents the proportion of pixels in the image with intensity values less than or equal to a given intensity level $x$. It provides a thorough summary of pixel intensities across the entire image. The ECDF value increases with rising $x$, spanning from $0$ (signifying that no pixels have values less than or equal to $x$) to 1 (showing that all pixels satisfy the criterion $x_{ji} \leq x$). The ECDF elucidates the distribution of pixel intensities, enabling the examination and comparison of image attributes such as brightness and contrast.

Let $\mathcal{F}_j(x)$ denote the ECDF of the pixel values from the $j$-th image, where $j = 1, 2, ..., N$. Upon calculating the ECDF for a collection of pictures, the DFDV consolidates the ECDF values at designated distinguishing points. The DFDV at a particular discriminating point $T$ is defined as the probability that the ECDF values at that point are less than or equal to a certain value $y$. The DFDV function at point $T$ can be articulated as:

$$G_T(y) = P\left(\left\{\mathcal{F}_j(x) \in \mathfrak{F} | \mathcal{F}_j(T) \leq y\right\}\right) \ \text{for } y \in [0, 1]$$

(18)

$\mathcal{F}_j(x)$ signifies the ECDF of the $j$-th image, whereas $\mathfrak{F}$ denotes the aggregation of ECDFs from all $N$ images. The term $T$ denotes a certain pixel intensity level utilised as a differentiating criterion. The variable $y$, which ranges from 0 to 1, represents the cumulative probability at $T$, namely the value of the ECDF assessed at point $T$ [25]. These distinguishing features encapsulate essential traits in the distribution of pixel intensities among various image categories. The discrepancies in the choice of these sites give different shape of features hence influence the discriminative ability of the features obtained. The creation of ECDF and the further conversion into the DFDV is a procedure described in [25] and provides a well-structured approach to represent the data of the image in symbolic form by using the cumulative distribution-based features.

The DFDV classifies different images according to the distribution of their pixels intensity. It finds important distribution patterns which distinguishes one type of photos over another by comparing DFDV at certain distinguishing points. The representation is more concezed symbolic data representation, unlike traditional pixel-based data. DFDV data is output data which is minimised and therefore computationally powerful whilst preserving the capability of portraying complex relationships existing between pixel intensities inside the visual image [38,39].

## Cluster-based point selection distinction

The point of distinction $T \in \{0, 1, 2, ..., 255\}$ is a particular value of a pixel which is used in the assessment of the ECDF at the point. The probability value $y \in [0, 1]$, or the realisation of the ECDF $\mathcal{F}_j(x)$ at point $T$ is also represented as $\mathcal{F}_j(T)$. DFDV function is given by $G_T(y)$ and it is the probability that $\mathcal{F}_j(T) \leq y$. This gives information on the detailed spread of pixel values in that area.

The K-means clustering algorithm detects the points of distinction. They assign the data $y_j$, which are shown as $\mathcal{F}_j(T)$, into $v$ cluster groups based on similarity [40,41]. It starts with the random choice of $v$ initial centroids, $c_1, c_2, \ldots, c_v$. We assign every piece of data $y_j$ to the cluster those with the nearest centroid, using the following measure of distance:

$$\mathbb{C}_j = \underset{k}{argmin}\, |y_j - c_k| \tag{19}$$

$$c_k = \frac{1}{N_k} \sum_{j=1}^{N_k} y_j \tag{20}$$

$\mathbb{C}_j$ represents the cluster assigned to $y_j$, whereas $c_k$ represents the centroid value of cluster $k$ for $k = 1, 2, \ldots, v$. Afterward, the recalibration of the centroids is performed by taking the mean of those values of the data points in every cluster [42]. $N_k$ is the number of data points of cluster $k$. The redistribution of the data points and the switching of the centroid is repeated until the convergence of the clusters, which means the convergence of the algorithm [43].

**Algorithm 1. K-means Clustering of ECDF points** $\mathcal{F}_j(T)$

```
Input: Set of ECDF values {y_j}, for j = 1, 2, ..., N, number of clusters v
Output: Cluster assignments ℂ_j, centroids c_k
1: Initialize v centroids randomly {c_1, c_2, ..., c_v}
2: Repeat:
    For each y_j in data:
        Assign y_j to cluster with nearest centroid
    For each cluster k:
        Update centroid c_k as mean of assigned points
3: Until centroids converge
```

In this study, the number of clusters was set to $v = 10$, corresponding to the ten digit classes (0–9) in the MNIST dataset. This design choice ensures that the clustering process aligns structurally with the class composition of the dataset. Euclidean distance was used as the similarity measure, and centroid updates were iterated until convergence.

This method clusters ECDF values to acquire the unique characteristics of pixel values distribution in each category of images that enable determination of the most suitable discriminative level to use in the classification. After grouping the data, the effectiveness of the obtained clusters can be evaluated using the Silhouette coefficient, which measures the effectiveness of the groups. Suppose that we have a dataset of $N$ points $y_j$ (where $j = 1, 2, \ldots, N$), referred to the value of ECDF at point $T$ for the $j$-th image.

The Silhouette coefficient for point $y_j$ is determined as follows:

$$s\left(y_j\right) = \frac{b\left(y_j\right) - a\left(y_j\right)}{maks\left(a\left(y_j\right), b\left(y_j\right)\right)} \tag{21}$$

$$a\left(y_j\right) = \frac{1}{n-1} \sum_{y_{j'} \in K, y_{j'} \neq y_j} d\left(y_j, y_{j'}\right) \tag{22}$$

$$b\left(y_j\right) = \frac{1}{n_{j'}} \sum_{w_k \in K(y_{j'})} d\left(y_j, w_k\right) \tag{23}$$

Let mean $a(y_j)$ to other data points $y_j$ be the average of all data points in the same cluster to $y_j$. Conversely, $b(y_j)$ denotes the mean distance from $y_j$ to points in the closest neighbouring cluster. To compute $a(y_j)$, assume cluster $K$ consists of n points that are part of the same cluster as $y_j$, where $d(y_j, y_{j'})$ denotes the distance between $y_j$ and $y_{j'}$. To calculate $b(y_j)$, denote $K(y_j)$ as the cluster including $y_j$ and $K(y_{j'})$ as the nearest neighbouring cluster. Let $\dot{n}_j$ and $\dot{n}_{j'}$ represent the amounts of points in clusters $K(y_j)$ and $K(y_{j'})$, respectively. $d(y_j, w_k)$ denotes the distance between point $y_i$ and $w_k$, situated in distinct clusters [44]. The overall Silhouette coefficient $\overline{S}$, for the entire dataset is the mean of all individual coefficients.

$$\overline{S} = \frac{1}{N} \sum_{j=1}^{N} s(y_j)$$

(24)

The Silhouette coefficient ranges from −1 to 1. A number close to 1 indicates that a data point is well-aligned with its cluster and distinctly separates from others. A score of about −1 indicates that the data point is closer to an alternative cluster, suggesting a potential clustering error [45]. The mean Silhouette coefficient for all clusters is calculated at various candidate points to determine the optimal distinguishing point that maximises the average Silhouette coefficient.

Distinguishing points are selected using a silhouette-based optimization strategy. First, ECDF $F_j(x)$ are constructed for all images. A set of $\dot{s}$ candidate threshold values $\tilde{x}_t$ is generated across the ECDF domain, and the corresponding ECDF values are computed. For each candidate threshold, K-means clustering (Algorithm 1) is performed with $K = 10$ clusters, consistent with the ten MNIST digit classes, using Euclidean distance. The average silhouette coefficient $\overline{S}_t$ is then calculated to assess clustering quality. The first distinguishing point is selected as the candidate maximizing $\overline{S}_t$, and the second is chosen as the next highest value excluding the first. In this study, the number of distinguishing points is set to $s = 2$, balancing clustering quality and model complexity within the copula framework. The overall procedure is illustrated in Fig 1.

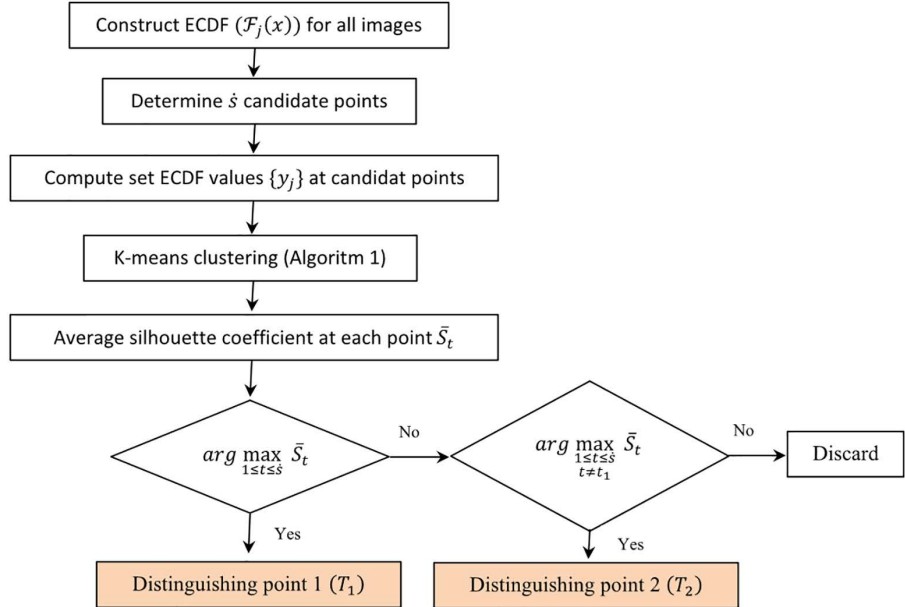

**Fig 1. Flowchart for selecting two optimal distinguishing points based on average Silhouette coefficients.** The first distinguishing point is selected as the candidate maximizing the average silhouette coefficient $\overline{\mathbf{S}}_t$, and the second is chosen as the next highest coefficient excluding the first selected point.

## Gaussian Copula with DFDV for image classification

This research employs a Gaussian copula methodology for image classification, utilising random variables characterised by the DFDV derived from the ECDF. The algorithm categorises photos into classes $k = 0, 1, \ldots, v - 1$, with each image $j = 1, 2, \ldots, N_k$ segmented into $l = 1, 2, \ldots, q$ areas. For each partition, the points $t = 1, 2, \ldots, s$ are utilised to assess pixel values, yielding $p = qs$ random variables. These random variables encapsulate the fundamental distribution patterns of pixel intensities throughout the image. Implementing this technique necessitates several essential processes. Each of these procedures is detailed in the subsequent sections.

The distinguishing points $T_t$ are identified by the analysis of comprehensive data from $v$ classes. The ECDF is initially created for $N$ data points across all classes, and $s$ candidate distinguishing points are chosen. These points signify critical thresholds that may elucidate disparities in the pixel intensity distributions among various classes. The ECDF for a particular class $k$, partition $l$, and image $j$ is represented as $\mathcal{F}_{klj}(x)$, adhering to Equation (17). This function offers a cumulative depiction of pixel intensities for a specified area of the image. The DFDV, originating from the ECDF, encapsulates the distribution at designated locations $T_t$, where $t = 1, 2, \ldots, s$. For partition $l$, and class $k$, the DFDV is denoted as $G_{k,l,t}(y_{k,l,t})$, as articulated in Equation (18). This transformation enables a concise symbolic representation of visual regions, hence enhancing subsequent analysis and classification.

The $k$-means algorithm is employed to partition the data into $v$ clusters for each distinguishing point. The clustering procedure is directed by Equations (19) and (20), which guarantee the establishment of significant clusters. Equations (21–23) calculate the Silhouette coefficient to assess the quality of clustering at each prospective distinguishing point. This statistic assesses the extent to which each data point aligns with its assigned cluster compared to other clusters. We determine the optimal threshold as the point that maximises the average Silhouette coefficient. This ensures that the chosen distinguishing point embodies the characteristics that distinguish image classes.

This study consistently develops the copula model for each class, enhancing generality. The copula model for the $k$-th class can be expressed as follows:

$$H\left(y_{k,1,1}, y_{k,1,2}, \ldots, y_{k,q,s}\right) = C\left(G_{k,1,1}\left(y_{k,1,1}\right), G_{k,1,2}\left(y_{k,1,2}\right), \ldots, G_{k,q,s}\left(y_{k,q,s}\right)\right) \tag{25}$$

This formulation enables the copula function $C$ to encapsulate the dependency structure among random variables, while the marginal distributions $G_{k,l,t}\left(y_{k,l,t}\right)$ manage the individual behaviour of each variable. Through this generalisation, the model is rendered applicable to a variety of image classes, thereby facilitating the consistent and systematic representation of the data. The random variables $\left(U_{k,1,1}, U_{k,1,2}, \ldots, U_{k,q,s}\right)$ are defined as transformations that satisfy the equation $U_{k,l,t} = G_{k,l,t}\left(Y_{k,l,t}\right)$, where the variables exhibit a uniform distribution over the interval $[0, 1]$. The correlation between these random variables is expressed as follows by a copula function:

$$H\left(y_{k,1,1,1}, y_{k,1,1,2}, \ldots, y_{k,d,q,s}\right) = C\left(u_{k,1,1,1}, u_{k,1,1,2}, \ldots, u_{k,d,q,s}\right) \tag{26}$$

Furthermore, the joint distribution function will be associated with the joint probability density function, which is defined as follows:

$$h\left(y_{k,1,1}, y_{k,1,2}, \ldots, y_{k,q,s}\right) = c\left(u_{k,1,1}, u_{k,1,2}, \ldots, u_{k,q,s}\right) \prod_{l=1}^{q} \prod_{t=1}^{s} g_{k,l,t}\left(y_{k,l,t}\right) \tag{27}$$

This study employed the Gaussian copula model to examine the interrelationship among random variables related to features, partitions, and distinguishing points within the image. The Gaussian copula function utilised in this work is articulated as follows:

$$C\left(G_{k,1,1}\left(y_{k,1,1}\right), G_{k,1,2}\left(y_{k,1,2}\right), \ldots, G_{k,q,s}\left(y_{k,q,s}\right) \mid \Lambda\right)$$
$$= \Phi_{\Lambda}\left(\Phi^{-1}\left(G_{k,1,1}\left(y_{k,1,1}\right)\right), \Phi^{-1}\left(G_{k,1,2}\left(y_{k,1,2}\right)\right), \ldots, \Phi^{-1}\left(G_{k,q,s}\left(y_{k,q,s}\right)\right)\right)$$

(28)

The density function of the Gaussian copula is obtained from its distribution function as illustrated below:

$$c\left(G_{k,1,1}\left(y_{k,1,1}\right), G_{k,1,2}\left(y_{k,1,2}\right), \ldots, G_{k,q,s}\left(y_{k,q,s}\right) \mid \Lambda\right) = \frac{1}{\sqrt{|\Lambda|}} exp\left\{-\frac{1}{2}z^T\left(\Lambda^{-1}-I_p\right)z\right\}$$

(29)

With $z = \left(\Phi^{-1}\left(G_{1,1,1}\left(y_{1,1,1}\right)\right), \Phi^{-1}\left(G_{1,1,2}\left(y_{1,1,2}\right)\right), \ldots, \Phi^{-1}\left(G_{d,q,s}\left(y_{d,q,s}\right)\right)\right)^T$. The joint density function of the Gaussian copula is expressed as follows:

$$h\left(y_{k,1,1}, y_{k,1,2}, \ldots, y_{k,q,s}\right) = \frac{1}{\sqrt{|\Lambda|}} exp\left\{-\frac{1}{2}z^T\left(\Lambda^{-1}-I_p\right)z\right\}\prod_{l=1}^{q}\prod_{t=1}^{s} g_{r,l,t}\left(y_{r,l,t}\right)$$

(30)

We come up with the likelihood function in order to simultaneously involve both the copula and marginal distributions in establishing the parameters of the Gaussian copula model. This joint likelihood formulation consists of two elements: one for the Gaussian copula and another for the marginal components, as follows:

$$\ell\left(\Theta, \Lambda \mid y_{k,1,1}, y_{k,1,2}, \ldots, y_{k,q,s}\right) = \ell_c\left(\hat{\Theta}, \Lambda \mid y_{k,1,1}, y_{k,1,2}, \ldots, y_{k,q,s}\right) + \sum_{j=1}^{N_k}\ell_{g_{r,l,t}}\left(\theta_{k,l,t} \mid y_{k,1,1,i}, y_{k,1,2,i}, \ldots, y_{k,l,t,i}\right)$$

(31)

Denote the parameters correlated with the marginal distributions of the copula by $\Theta = \left[\theta_{k,1,1,1}, \theta_{k,1,1,2}, \ldots, \theta_{k,d,q,s}\right]$ the log-likelihood of the Gaussian component of the copula is $\ell_c$. The second in the summation symbolizes the log-likelihood of the marginals distributions whereby; $\ell_{g_{r,l,t}}$ denotes the log-likelihood that is related to the marginal distributions of every territory and division.

Parameter estimation is performed by maximizing the joint log-likelihood function with respect to both $\Theta$ and $\Lambda$. To ensure computational efficiency and numerical stability, the optimization is conducted using the two-step Inference Functions for Margins (IFM) procedure described in Equations (12) and (13). In the first stage, the marginal parameters $\Theta$ are estimated independently by maximizing the marginal likelihood function as given in Equation (12). Subsequently, the copula parameter $\Lambda$ is estimated by maximizing the copula log-likelihood in Equation (13), while keeping the marginal estimates fixed. This procedure enables coherent modeling of both the marginal distributions and their dependence structure within the Gaussian copula framework.

## Classification procedure

The method used in this study is one of classification, but this is done using a probabilistic framework where the intention is to determine the likelihood of a particular test image being related to a particular class. Let $\dot{X}$ and $C_k$ be the new input image and the $k$-th class respectively. The conditional probability $P\left(C_k \mid \dot{X}\right)$ shows the assigned probability that an input falls in an appropriate $C_k$ class. It is computed by first calculating the joint probability of the input and the class and this is done by multiplying two factors namely: (1) probability of observing $X$ centering dot given the probability of class $C_k$ which is expressed as $P\left(\dot{X} \mid C_k\right)$, and (2) the prior probability of class $C_k$ which has the probability expressed as $P\left(C_k\right)$. Probability $P\left(\dot{X} \mid C_k\right)$ is estimated using a Gaussian copula model trained on the data so far, but prior $P\left(C_k\right)$ is calculated as training samples belonging to the $k$-th class divided by the entire trained sample set [46].

After computing the conditional probabilities of each of the classes, the machine will assign the test image to the one with the greatest probability. This approach allows making classification judgements using the most dependable

statistical records extracted out of the data and model. The confusion matrix evaluates the effectiveness of the classification model that counts and lists all correct and incorrect guesses of the model on each of the classes. The main performance indicator is the accuracy which refers to the number of successful predictions made to the total amount of forecasts. Moreover, precision rate, recall and F1-score are also obtained to have a more comprehensive evaluation of the success of this model. These measures are used together to test the competency of the model in distinguishing and disparity between classes of images to guarantee overall analysis of the accuracy of the model in prediction [47]. The overall structure of the proposed approach, including both training and testing phases, is presented in Fig 2.

Fig 2 illustrates the overall classification framework. In the training phase, images are partitioned and their ECDFs are constructed. Optimal distinguishing points are selected based on the highest silhouette coefficients (see Fig 1). Using ECDF values at these points, the parameters of the DFDV $G_T(y)$, denoted by $\hat{\theta}$, are estimated for each class. The dependence structure among features is then modeled using a Gaussian copula with correlation parameter $\hat{\Lambda}$. In the testing phase, ECDF values at the selected distinguishing points are evaluated using the estimated parameters $\hat{\theta}$, and the joint copula density $h(y)$ is computed using $\hat{\Lambda}$. Each test image is assigned to the class that maximizes the corresponding joint density value.

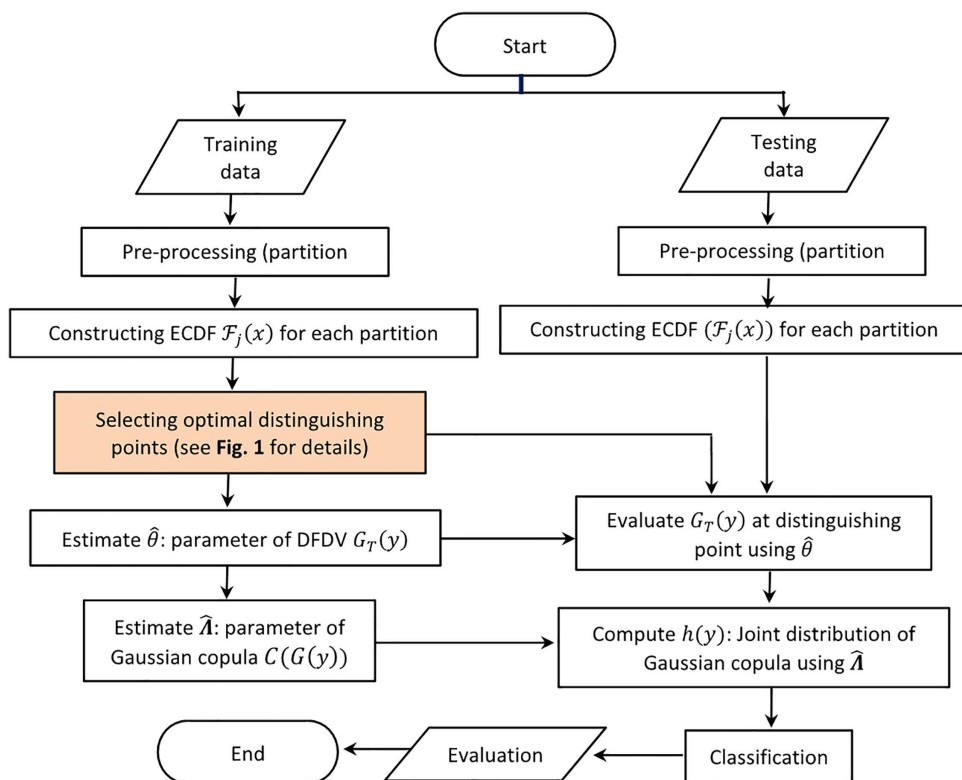

**Fig 2. Block diagram of the proposed image classification method using symbolic ECDF and DFDV features.** Starting from data preprocessing and ECDF construction on partitioned images. Distinguishing points are selected based on clustering ECDF values and evaluating Silhouette coefficients (detailed in Fig 1). The DFDV $G_T(y)$ and Gaussian copula parameter $\hat{\Lambda}$ are estimated from training data. In testing, the model evaluates $G_T(y)$ and computes the joint copula distribution $h(y)$ for final classification.

## Experiment and results

In order to decide upon the effectiveness of the given approach, one conducted experiments to analyze a well-known unofficial benchmark in image classification, MNIST dataset. The dataset contains 70,000 grayscale images of digital handwritings (0–9) with a size of 28 × 28 pixels each, and is divided into 60,000 training and 10,000 test set. Every image is in the form of a matrix that comprises pixel intensity values between 0 and 255. As a part of the present study, symbolic data denoted by Gaussian copula modelling is used to improve classification. The pixel intensity should be normalised and features should be extracted using the ECDF and the DFDV as the preprocessing activities. DFDV is a random variable that should summarise critical distributional assumptions of pixel intensity values over image partitions. We estimated the model of the Gaussian Copula by applying the methods explained in the previous section. The performance of the classification was measured by using the mean accuracy as well as the maximum accuracy of the test set.

### Preprocessing data

This experiment preprocesses the handwritten numeric dataset MNIST to maximise on the feature extraction to increase the performance of the classification accuracy. The dataset is freely available at https://www.kaggle.com/datasets/hojjatk/mnist-dataset. All of the images are rescaled to a 20 × 20 grid, which is enough to memorise the basic outline of the numeric shapes and reduces excess information [48]. Each of the enlarged pictures is partitioned into equal sections to better reflect characteristics of the photos. This decomposition allows the study of localised properties and these are necessary in symbolic representations of data. The results of the previous studies [25] showed that the best modelling results were achieved after dividing the images into four horizontal parts as can be seen in Fig 3.

This finding highlights the need of determining an optimal size of partition to achieve effective representation of distribution-related features during image classification. The pixel values are normalised using the min-max scaling approach a process in which after normalisation, the values adopt a range of [0,1] to normalise the data hence making manipulation possible.

### Outcomes of the distance points

The identifications of distinguishing points entail application of special pixel values that can best distinguish the features of classes. Clustering is executed for each of $\dot{s}$ candidate points (with $\dot{s} = 10$), ECDF values are clustered using K-Means. The Silhouette coefficient is computed to evaluate clustering quality. This coefficient assesses the degree to which each data point corresponds with its respective cluster in relation to other clusters. We identify the two points with the highest Silhouette coefficient as the distinguishing point, as it effectively differentiates the characteristics of many classes. Selecting 10 candidate points ensures adequate coverage of pixel variations while maintaining computational efficiency. Limiting the final selection to two distinguishing points reduces model complexity and mitigates the risk of overfitting. Fig 4 depicts the methodology, including the methods for pinpointing these locations. We produced the ECDF for all images spanning the ten classes.

We select two distinguishing points for each partition. The resulting points may vary for each occurrence, depending on the value of the acquired Silhouette coefficient. The chosen distinguishing qualities are those that most efficiently differentiate the image characteristics among its categories. We construct the Gaussian copula model utilising the IFM approach and optimal distinguishing points. Table 1 presents the mean Silhouette coefficient values.

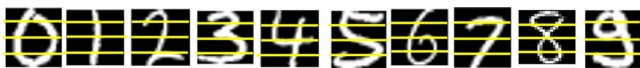

**Fig 3. Image partitioning into four horizontal segments.** Each digit image is divided into four horizontal parts to extract localized ECDF features.

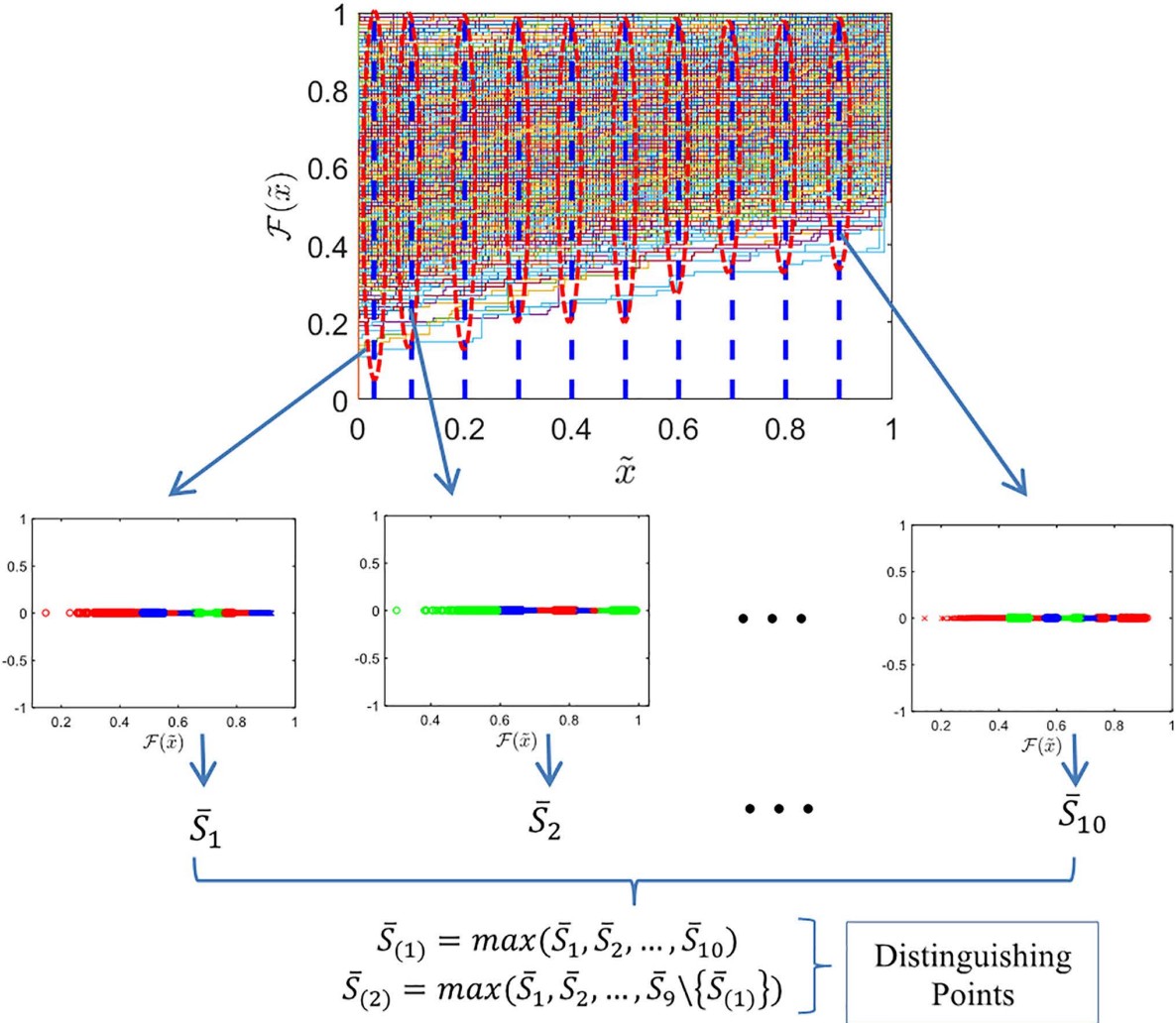

**Fig 4. Illustration of distinguishing point determination selection using K-means clustering and Silhouette coefficients.** The first step is constructing the ECDF for all **N** data. ECDF values $F_j(x)$ are evaluated at **t** candidate thresholds, where in this case **t = 10**. At each candidate point, the ECDF values are clustered into **v** groups using K-means clustering, and the clustering quality is assessed using the average Silhouette coefficient $\bar{S}_t$. the two thresholds with the highest $\bar{S}$ values are selected as optimal distinguishing points..

**Table 1. Silhouette coefficient values for 10 candidate distinguishing points.** Each row corresponds to a specific image partition *l* = 1, 2, 3, 4, and each column represents a candidate distinguishing point $T_t$ with $\tilde{x} \in \{0.03, 0.1, \ldots, 0.9\}$. For each partition, the two candidate points with the highest Silhouette coefficient values (in bold) are selected as optimal distinguishing points.

| Partition | $T_t$ with t= 1, 2, ..., 10 | | | | | | | | | |
|---|---|---|---|---|---|---|---|---|---|---|
| | $\tilde{x}$= 0.03 | $\tilde{x}$= 0.1 | $\tilde{x}$= 0.2 | $\tilde{x}$= 0.3 | $\tilde{x}$= 0.4 | $\tilde{x}$= 0.5 | $\tilde{x}$= 0.6 | $\tilde{x}$= 0.7 | $\tilde{x}$= 0.8 | $\tilde{x}$= 0.9 |
| 1 | 0.6596 | 0.6611 | 0.6440 | 0.6559 | 0.6789 | 0.6882 | 0.6614 | 0.6880 | 0.6897 | 0.6804 |
| 2 | 0.6863 | 0.6620 | 0.6711 | 0.6599 | 0.6773 | 0.6839 | 0.6749 | 0.6670 | 0.6795 | 0.6663 |
| 3 | 0.6835 | 0.6749 | 0.6847 | 0.6923 | 0.6835 | 0.6929 | 0.6889 | 0.6703 | 0.6702 | 0.6710 |
| 4 | 0.6761 | 0.6842 | 0.6811 | 0.6742 | 0.6858 | 0.6566 | 0.6530 | 0.6846 | 0.6423 | 0.6751 |

For each $T_t$, the pixel thresholds $(\widetilde{x})$ associated with the two highest Silhouette coefficients are selected as distinguishing points. This guarantees that the chosen points yield optimal separability for clustering among classes. At partition 1, the maximum Silhouette value is recorded at $\widetilde{x} = 0.8$ with a Silhouette coefficient of 0.6897, indicating that this threshold yields the most distinct clustering for this scenario. The distinguishing points for each threshold $(T_t)$ are selected based on the two highest Silhouette values to provide excellent separability. In partition 1, the positions $\widetilde{x} = 0.5$ and $\widetilde{x} = 0.8$ demonstrate the highest silhouette values. Likewise, for partition 2, the chosen points are $\widetilde{x} = 0.03$ and $\widetilde{x} = 0.5$. Transitioning to partition 3, the best points are $\widetilde{x} = 0.3$ and $\widetilde{x} = 0.5$. Finally, for partition 4, the points $\widetilde{x} = 0.4$ and $\widetilde{x} = 0.7$ have the optimal clustering performance. These values serve as determining thresholds that distinctively separate the classes of images to be analyzed later on. The selected distinctive points represent pixel value which provides the most reliable feature of separation between image classes. These can provide points to use as the basis of the ensuing clustering or classification in the pipeline of the analysis.

## Results for ECDF and DFDV

After identifying distinguishing features on the basis of the greatest Silhouette coefficients, there is the need to calculate the partitions and classes ECDF, respectively. Each pixel intensity distribution of a class in each partition produces an ECDF, so there are four ECDFs per a class, as there are four partitions. For each division, two distinguishing points are chosen, resulting in eight distinguishing points for each class. These ECDFs illustrate the cumulative probability of pixel intensities, reflecting variations in image attributes across distinct partitions. Figs 5 and 6 illustrates the ECDFs for each class and partition, offering a visual depiction of the pixel distributions that underpin the development of the Gaussian copula model.

ECDF plots represent full distribution of pixel brightnesses in the separate image divisions (rows) belonging to different classes of images (columns). The visualisation comprises two segments: the upper segment presents the ECDF results for classes 0–4, whereas the lower segment exhibits the ECDF results for classes 5–9. Every row is a representation of an image partition and every column is a representation of image classes numbered 0–9. The distribution patterns in the graphs underscore discrepancies among partitions and classes. In certain classes, pixel distributions are dispersed throughout the intensity spectrum, signifying variability in pixel attributes within those segments. Conversely, other classes have more concentrated distributions within particular intensity ranges, indicating a predominance of pixels with uniform attributes. This phenomenon is seen in specific partitions when the ECDF curves ascend sharply over a limited range, indicating that a significant proportion of pixels have the same intensity value.

The disparities in distribution patterns among classes and partitions indicate that each class exhibits distinct traits as evidenced by its pixel distributions. The divisions segment the image, facilitating a more nuanced study of pixel changes. The red and blue vertical lines in the plots serve as essential markers for comprehending the data distribution and constructing the Gaussian copula model.

The development of DFDV relies on the collection of ECDFs at the specified distinguishing points indicated in the plots. For each image class, a total of 8 DFDVs are produced at the distinguishing points for each partition. The DFDVs will be used as random variables in Gaussian model of copula. Illustration of the DFDV results is depicted in Fig 7.

According to the DFDV plot shown above, it is observed that the DFDV distribution pattern is various in each image class ($k = 0$ to $k = 9$). The curves of DFDV of each of the classes differ in their shape and in the points of distribution with higher concentrations attesting to the differences of their type. The DFDV distribution is more cluster in some classes, e.g., $k = 1$ and $k = 4$ while it is more dispersed with flatter peaks in other classes, e.g., $k = 0$ and $k = 5$. This trend shows that the DFDV values differ among the image class and partition as a difference between the classes in the pixel intensity characteristic. The divergence in DFDV distribution contributes as an influential variable in comprehending the disparity in pixel features of each partition and forms the input in the Gaussian copula prototyping to better simulate the associations between partitions. These findings also reveal the suitability of DFDV in identifying the slight changes in distributions of pixel intensities.

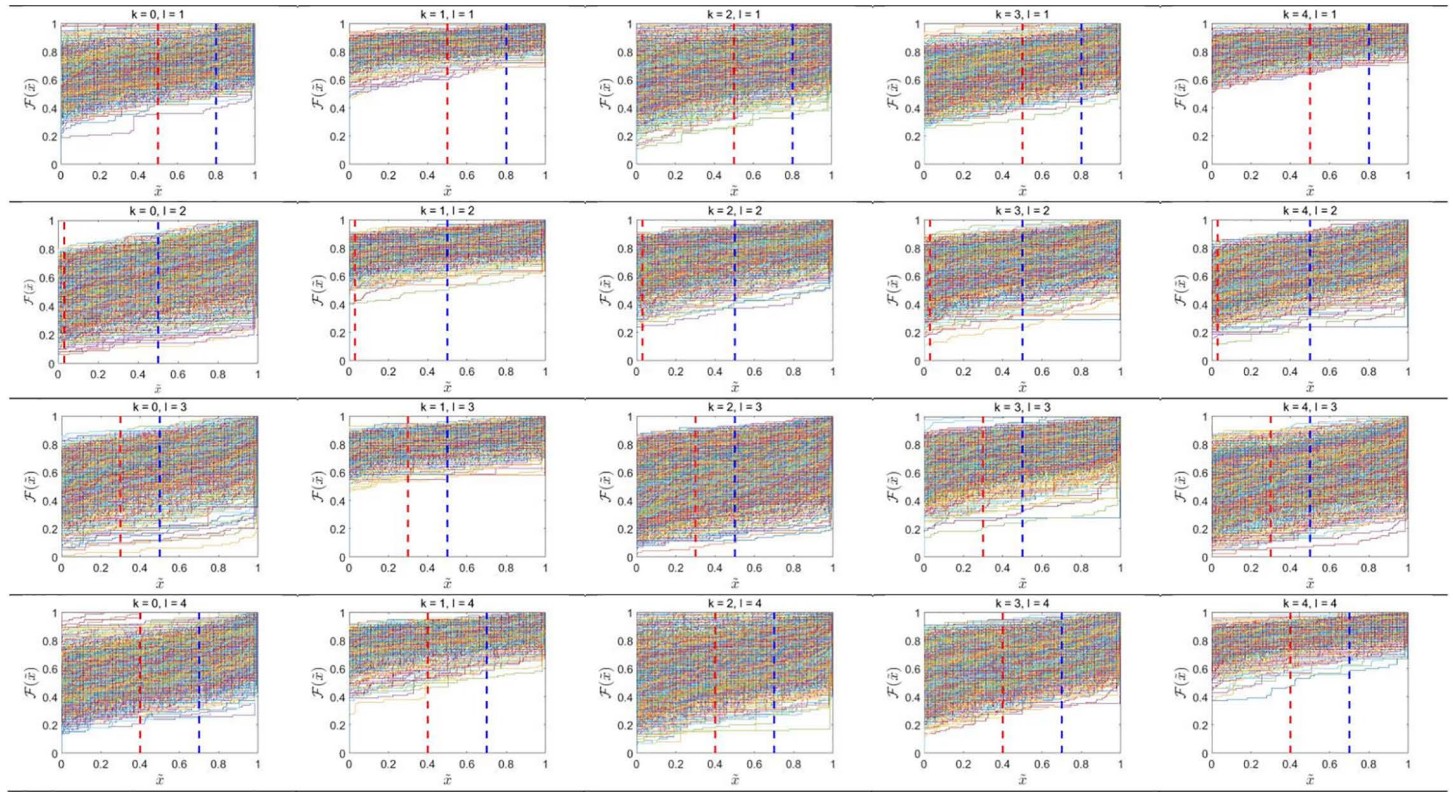

**Fig 5. ECDF plots of four partitions for digit classes k = 0, 1, ..., 4.** Each subplot shows the ECDF $\mathbf{F}(\tilde{x})$ with rows corresponding to partitions l = 1, 2, 3, 4 and columns to digit classes. Red and blue vertical dashed lines denote the first ($T_1$) and second ($\mathbf{T}_2$) selected distinguishing points, respectively.

The graphic of DFDV presented above shows the discrete patterns of DFDV distributions applying to each class of images ($k$ = 0 through to $k$ = 9). Each of the classes represents a certain level of variance in DFDVs shape and in the locations of the peaks, resulting in the individual profile of classes. In some of the classes, like variation in shape and distribution peak points, it is possible to present the variations in the characteristics of classes. The distribution of the DFDV in certain classes, e.g., $k$ and $k$ = 4 is clustered where the peaks are pronounced whereas in certain other classes, e.g., $k$ = 0 and $k$ = 5, the distribution is more spread with the peaks being low. This trend indicates that the DFDV values change depending on the image class and the partition by depicting variations in the intensity of pixels between the classes. The match inequalities in DFDV patterns are an important indicator to understand the deviation in pixel features in thousands of classes and is used as an input in Gaussian copula model to better describe the relationship between partitions. These findings also depict the sensitivity of DFDV to subtle variations in pixel intensity distributions.

## Results for Gaussian Copula Model

After obtaining the values of the DFDV at the characteristics points, the next stage involves analysis of distributional properties of the DFDV. KDE method is applied as a non-parametric way of accurately and smooth drawing probability density function of the DFDV to help understand the distributional properties. KDE enables the evaluation and comparison of the DFDV distribution for each image class without necessitating any prior assumptions on the distribution's structure. The estimation of the KDE parameter primarily involves determining the bandwidth, which regulates the smoothness of

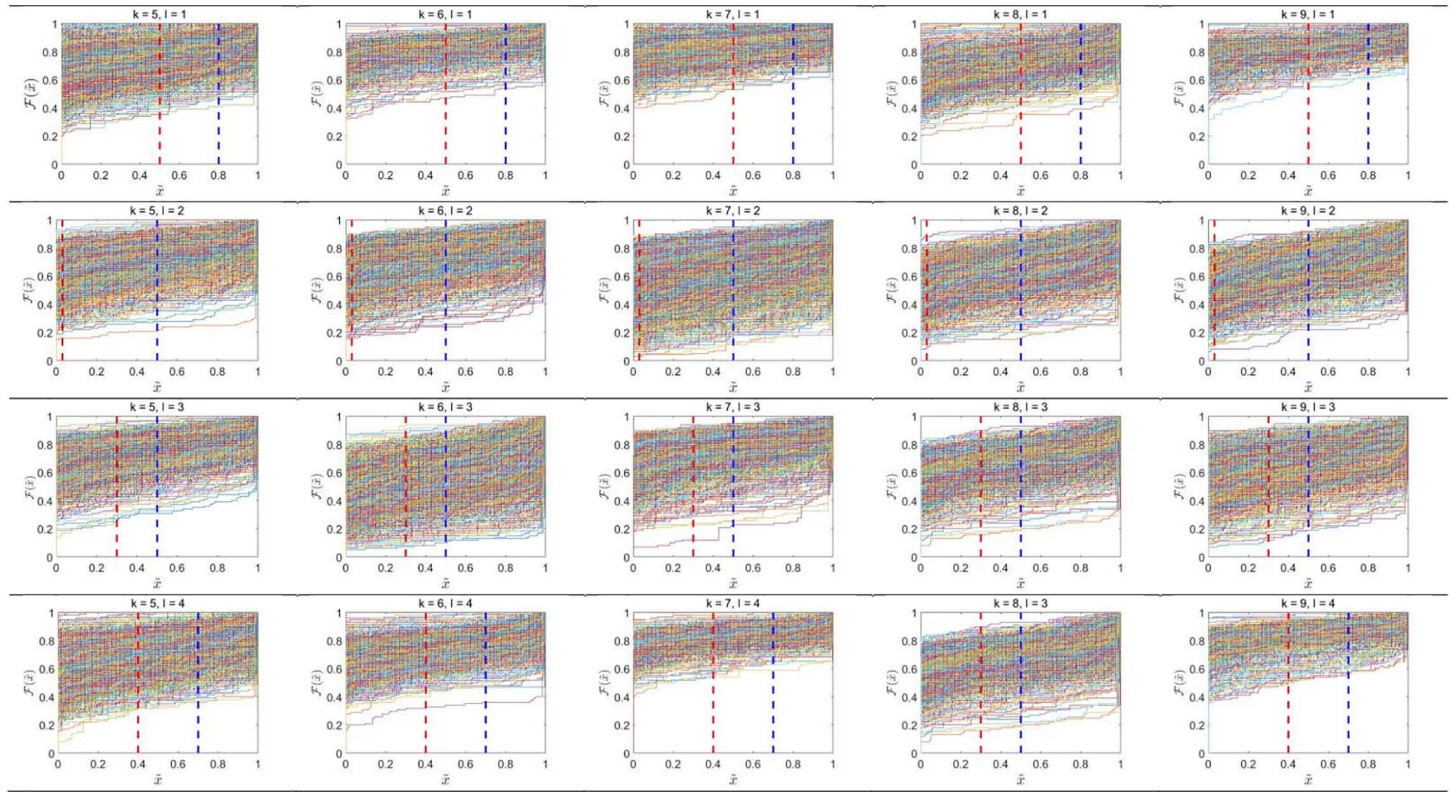

**Fig 6. ECDF plots of four partitions for digit classes k = 5, 6, ..., 9.** Each subplot shows the ECDF $\mathbf{F}(\tilde{x})$ with rows corresponding to partitions l = 1, 2, 3, 4 and columns to digit classes. Red and blue vertical dashed lines denote the first ($T_1$) and second ($T_2$) selected distinguishing points, respectively.

the computed density. Equation (16) estimates the parameter, while the Kolmogorov-Smirnov (KS) test assesses the derived bandwidth values for goodness-of-fit. The KS test quantitatively evaluates the degree of correspondence between the DFDV data and the distribution generated by KDE. Table 2 encapsulates the expected bandwidth parameters and the outcomes of the KS test.

This analysis assesses the bandwidth of KDE and the $P_{values}$ from the KS test over different partitions and thresholds ($T_1$ and $T_2$). The minimal bandwidth values ($0.01 - 0.03$) signify narrow kernel smoothing. $P_{values} \geq \alpha$ signify a strong correspondence between the ECDF and KDE distributions. The copula model employs a Gaussian copula approach to define the joint distribution of eight interrelated random variables, denoted as $y_{k,1,1}, y_{k,1,2}, \ldots, y_{1,4,2}$. The copula density function is intended to represent both the marginal behaviours of the variables and their dependence structure. The model is defined as:

$$h\left(y_{k,1,1}, y_{k,1,2}, \ldots, y_{1,4,2}\right) = \frac{1}{\sqrt{|\mathbf{\Lambda}|}} exp\left\{-\frac{1}{2}\mathbf{z}^T\left(\mathbf{\Lambda}^{-1} - I_8\right)\mathbf{z}\right\} \prod_{l=1}^{4}\prod_{t=1}^{2} g_{k,l,t}\left(y_{k,l,t}\right)$$

(31)

Let $\mathbf{z} = \left(\Phi^{-1}\left(G_{k,1,1,1}\left(y_{k,1,1,1}\right)\right), \ldots, \Phi^{-1}\left(G_{k,1,4,2}\left(y_{k,1,4,2}\right)\right)\right)^T$ and $\mathbf{\Lambda}$ denotes the covariance matrix illustrating the interrelationships among the eight random variables. $G$ signifies the marginal cumulative distribution function of each variable,

whereas $\Phi^{-1}$ symbolises the inverse of the ordinary normal cumulative distribution function. The terms $g_{k,l,t}\left(y_{k,l,t}\right)$ represent the marginal density functions of the respective variables.

The parameters of the Gaussian copula model are determined by formulating a likelihood function based on the copula density function. This procedure incorporates the dependence structure represented by the correlation matrix $\Lambda$ and the marginal distributions $g_{k,l,t}\left(y_{k,l,t}\right)$. Parameter estimation is conducted via the IFM, facilitating a distinct and efficient delineation of the marginal components and dependencies, hence yielding optimal parameter estimations. The estimated parameters are displayed in Table 3.

We performed clasfsification utilising the Bayesian technique, grounded in the predicted parameter values of the obtained covariance matrices. Following this approach, the data are grouped into the right categories by using the information regarding the interdependence of various variables which people have identified in the covariance matrices. Performance measures on the classification results were then carried out to estimate the aptitude of the model to categorize the categories accurately. The metrics used in the analysis will consist of accuracy, precision, recall and F1-score which will be presented in Table 4. A comparison is also made with the previous approach [25] that employed fixed distinguishing points.

As shown in Table 4, the proposed method, which employs optimally selected distinguishing points based on the highest Silhouette coefficients, achieves an average accuracy of 68.27% and recall of 66.70%, outperforming the previous method that used fixed distinguishing points (62.22% and 62.46%, respectively). This improvement indicates better generalization and enhanced ability to distinguish class-specific distributional features across partitions. Although the previous approach achieves slightly higher maximum values in accuracy (96.92%) and precision (95.30%), the proposed method maintains competitive maximum performance (up to 95.40% recall), while demonstrating more consistent results overall. These findings affirm the robustness of the model and highlight the effectiveness of using optimized symbolic features in enhancing the classification of handwritten digits. To provide a more detailed view of the classification performance, the confusion matrix for the proposed method is presented in Fig 8.

## Discussion

The MNIST dataset of handwritten digits shows the relation between the pixel intensity distribution and the classification effectiveness through using the symbolic representation of data and the copula-based modeling. This is because preprocessing method that involves scaling of photos into a 20 times 20 grid and dividing the photos into four horizontal sections is indicative of the importance of local feature extraction of symbolic data analysis. This segmentation is precise in defining structural characteristics of hand written digits. Min-max scaling ensures that the input is standardised thus enhancing the reliability of the resultant analysis. The decidedly proper division size matches the previous research results [25].

Silhouette coefficient is necessary to achieve successful differentiation of classes of images. The two points on each segment have been selected so that they balance simplicity of the models against classification accuracy. Such results confirm the importance of selecting particular areas to reduce the dimensionality of data and preserve essential elements of the data to be correctly classified. The effectiveness of the clustering-based methodology is reflected by a comparative analysis with previous research. The same distinguishing points that were preset in the previous research seemed to have an average accuracy rate of only 62.44 percent [25]. This research also improved the mean accuracy of 68.27 percent through the identification of distinctive features through cluster analysis, clearly confirming that the proposed research methodology is useful in enhancing the performance of classifications. Moreover, the best precision increased up to 95.33 percent. In parallel to the accuracy, such metrics as precision, recall, and F1-score also demonstrated increases in relation to previous studies. It means that it is necessary to use cluster analysis to identify distinctive factors to improve the overall results of the model.

ECDF analysis is an elaborate analysis of the pixel intensity distribution in each of the section and category. Such different patterns on the ECDF curves mean that each class has its own peculiar properties because it shows differences

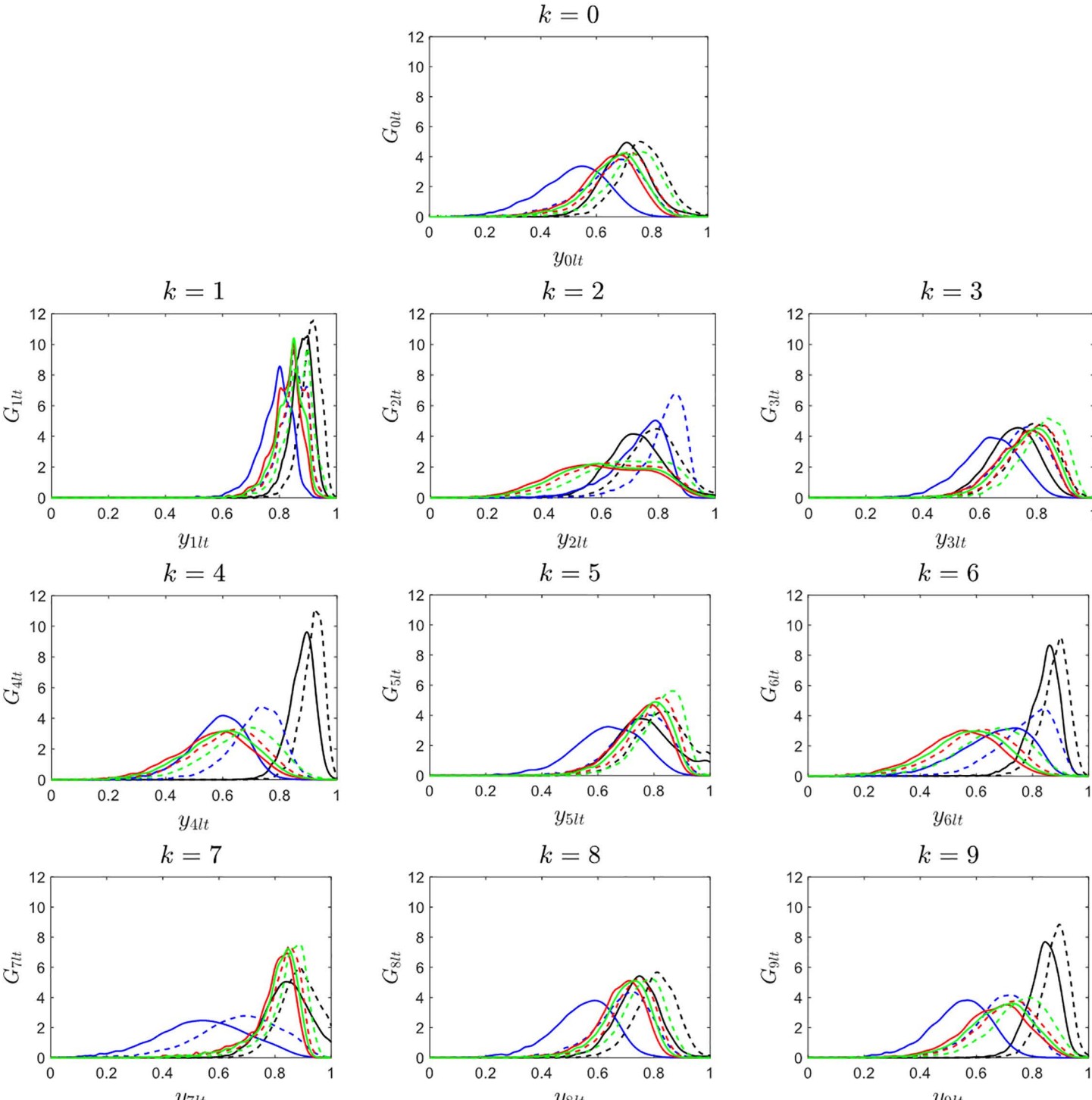

**Fig 7. DFDV results for ten image classes.** Each subplot corresponds to a digit class **k** = 0, 1, . . . , 9, showing the estimated DFDV curves **G**$_{klt}(y)$ for each of the 8 features formed from 4 partitions and 2 selected distinguishing points. The variations in curve shapes across digit classes reflect how distributional differences are captured and used for classification.

**Table 2. Estimated Bandwidth Parameters and Results of the Kolmogorov-Smirnov Test.** The table presents the bandwidth values (outside the parentheses) alongside the associated $P_{value}$ (inside the parentheses) for classes 0–9 over four partitions.

| Class | Partition 1 | | Partition 2 | | Partition 3 | | Partition 4 | |
|---|---|---|---|---|---|---|---|---|
| | $T_1$ | $T_2$ | $T_1$ | $T_2$ | $T_1$ | $T_2$ | $T_1$ | $T_2$ |
| 0 | 0.022 (0.052) | 0.019 (0.048) | 0.043 (0.053) | 0.031 (0.120) | 0.028 (0.090) | 0.025 (0.071) | 0.031 (0.093) | 0.028 (0.098) |
| 1 | 0.006 (0.077) | 0.006 (0.081) | 0.013 (0.073) | 0.010 (0.065) | 0.010 (0.074) | 0.010 (0.099) | 0.010 (0.108) | 0.009 (0.093) |
| 2 | 0.024 (0.081) | 0.025 (0.100) | 0.021 (0.064) | 0.013 (0.064) | 0.058 (0.088) | 0.050 (0.063) | 0.036 (0.099) | 0.029 (0.091) |
| 3 | 0.022 (0.064) | 0.018 (0.085) | 0.028 (0.130) | 0.021 (0.058) | 0.023 (0.054) | 0.020 (0.061) | 0.031 (0.064) | 0.025 (0.056) |
| 4 | 0.009 (0.080) | 0.006 (0.056) | 0.030 (0.110) | 0.022 (0.080) | 0.041 (0.071) | 0.037 (0.064) | 0.009 (0.083) | 0.009 (0.055) |
| 5 | 0.027 (0.094) | 0.021 (0.072) | 0.037 (0.130) | 0.025 (0.053) | 0.021 (0.072) | 0.018 (0.093) | 0.030 (0.080) | 0.024 (0.082) |
| 6 | 0.010 (0.069) | 0.010 (0.075) | 0.034 (0.092) | 0.024 (0.074) | 0.043 (0.058) | 0.039 (0.052) | 0.018 (0.064) | 0.017 (0.061) |
| 7 | 0.017 (0.062) | 0.015 (0.059) | 0.054 (0.082) | 0.039 (0.068) | 0.013 (0.068) | 0.013 (0.076) | 0.010 (0.080) | 0.009 (0.073) |
| 8 | 0.019 (0.063) | 0.016 (0.065) | 0.033 (0.084) | 0.025 (0.120) | 0.021 (0.058) | 0.020 (0.071) | 0.020 (0.090) | 0.019 (0.057) |
| 9 | 0.013 (0.099) | 0.009 (0.085) | 0.033 (0.082) | 0.025 (0.140) | 0.031 (0.072) | 0.028 (0.073) | 0.010 (0.072) | 0.009 (0.051) |

in the pixel distribution. ECDFs provide the DFDV values that highlight the differences between classes (with high peaks in certain classes (e.g., 1 and 4) and lower distributions in others (e.g., 0 and 5)). Such trends explain the effectiveness of DFDV in identifying subtle differences in the attributes of pixels.

The ability of Gaussian copula to describe the joint distribution of the DFDV over multiple regions emphasises the effectiveness of copula to establish the correlation among the local variables. The use of KDE in estimation of marginal distributions offers higher level of convenience in controlling non-standard distributions whose bandwidth parameter would be determined by examining the Kolmogorov-Smirnov test. Despite its advantages, the current study has certain limitations: the given reliance on rigid division methods can limit the flexibility of the system when handling a more complex range of data. However, the findings demonstrate the effectiveness of symbolic representation of data and the use of copulas modelling in classifications of images. The approach, particularly, suits the datasets with complex patterns of interconnection, as the case with MNIST application demonstrates.

The effectiveness of classification in this study, which integrates symbolic data and Gaussian copula on the MNIST dataset, is comparable with CNN-based applications, including LeNet, which achieve higher accuracies [49,50]. Our method also shows similar performance to traditional probabilistic models, including Bayesian and Gaussian Mixture Models, with mean accuracy around 75–85% [39]. Importantly, unlike CNNs or LeNet, our approach is interpretable and avoids the "black-box" issue. This transparency allows for better understanding of feature interactions and decision-making in classification tasks, which is often not possible with deep learning models. Although CNNs and LeNet achieve slightly higher accuracy, our method provides competitive performance while offering explainable results. These findings are also compatible with recent works using copulas in image classification [20,21,51] which demonstrate that copulas can effectively model the interactions among variables in visual datasets.

Although our method provides interpretable results, its accuracy is slightly lower than CNN and LeNet. The performance also depends on the choice of distinguishing points in the symbolic representation; different selections of these points can lead to varying classification results. Additionally, the current study focuses on the MNIST dataset, and further work is needed to assess scalability, robustness to noise, and generalization to more complex datasets. Future research could explore hybrid approaches that combine the interpretability of copula-based methods with the high accuracy of deep learning, as well as methods to optimize the selection of distinguishing points for improved performance.

**Table 3. Results of the estimated covariance matrix of the Gaussian Copula Model for digit classes $k = 0, 1, \ldots, 9$.** This table presents the estimated $8 \times 8$ covariance matrices $\hat{\Lambda}$ for the Gaussian copula model across all ten digits classes. Each matrix captures the dependence structure among the eight DFDV features derived from ECDF values at selected distinguishing points. Variations in the correlation strengths across classes reflect class-specific inter-feature dependencies used in the final classification stage.

| Class | Covariance Matrix | Class | Covariance Matrix |
|---|---|---|---|
| 0 | $\begin{bmatrix} 1 & 0.952 & 0.283 & 0.322 & \cdots & 0.760 \\ 0.952 & 1 & 0.385 & 0.425 & \cdots & 0.791 \\ 0.283 & 0.385 & 1 & 0.959 & \cdots & 0.553 \\ 0.322 & 0.425 & 0.959 & 1 & \cdots & 0.591 \\ \vdots & \vdots & \vdots & \vdots & \ddots & \vdots \\ 0.760 & 0.791 & 0.553 & 0.591 & \cdots\cdots & 1 \end{bmatrix}_{8\times8}$ | 5 | $\begin{bmatrix} 1 & 0.961 & 0.058 & 0.120 & \cdots & 0.733 \\ 0.961 & 1 & 0.093 & 0.159 & \cdots & 0.754 \\ 0.058 & 0.093 & 1 & 0.950 & \cdots & 0.344 \\ 0.120 & 0.159 & 0.950 & 1 & \cdots & 0.422 \\ \vdots & \vdots & \vdots & \vdots & \ddots & \vdots \\ 0.733 & 0.754 & 0.344 & 0.422 & \cdots\cdots & 1 \end{bmatrix}_{8\times8}$ |
| 1 | $\begin{bmatrix} 1 & 0.876 & 0.727 & 0.759 & \cdots & 0.833 \\ 0.876 & 1 & 0.731 & 0.757 & \cdots & 0.825 \\ 0.727 & 0.731 & 1 & 0.896 & \cdots & 0.739 \\ 0.759 & 0.757 & 0.896 & 1 & \cdots & 0.748 \\ \vdots & \vdots & \vdots & \vdots & \ddots & \vdots \\ 0.833 & 0.825 & 0.739 & 0.748 & \cdots\cdots & 1 \end{bmatrix}_{8\times8}$ | 6 | $\begin{bmatrix} 1 & 0.936 & 0.317 & 0.386 & \cdots & 0.758 \\ 0.936 & 1 & 0.346 & 0.422 & \cdots & 0.774 \\ 0.317 & 0.346 & 1 & 0.963 & \cdots & 0.461 \\ 0.386 & 0.422 & 0.963 & 1 & \cdots & 0.514 \\ \vdots & \vdots & \vdots & \vdots & \ddots & \vdots \\ 0.758 & 0.774 & 0.461 & 0.514 & \cdots\cdots & 1 \end{bmatrix}_{8\times8}$ |
| 2 | $\begin{bmatrix} 1 & 0.961 & 0.286 & 0.331 & \cdots & 0.805 \\ 0.961 & 1 & 0.338 & 0.387 & \cdots & 0.817 \\ 0.286 & 0.338 & 1 & 0.922 & \cdots & 0.275 \\ 0.331 & 0.387 & 0.922 & 1 & \cdots & 0.344 \\ \vdots & \vdots & \vdots & \vdots & \ddots & \vdots \\ 0.805 & 0.817 & 0.275 & 0.344 & \cdots\cdots & 1 \end{bmatrix}_{8\times8}$ | 7 | $\begin{bmatrix} 1 & 0.939 & 0.355 & 0.317 & \cdots & 0.479 \\ 0.939 & 1 & 0.286 & 0.246 & \cdots & 0.522 \\ 0.355 & 0.286 & 1 & 0.968 & \cdots & 0.402 \\ 0.317 & 0.246 & 0.968 & 1 & \cdots & 0.456 \\ \vdots & \vdots & \vdots & \vdots & \ddots & \vdots \\ 0.479 & 0.522 & 0.402 & 0.456 & \cdots\cdots & 1 \end{bmatrix}_{8\times8}$ |
| 3 | $\begin{bmatrix} 1 & 0.938 & 0.458 & 0.491 & \cdots & 0.710 \\ 0.938 & 1 & 0.501 & 0.543 & \cdots & 0.718 \\ 0.458 & 0.501 & 1 & 0.949 & \cdots & 0.571 \\ 0.491 & 0.543 & 0.949 & 1 & \cdots & 0.334 \\ \vdots & \vdots & \vdots & \vdots & \ddots & \vdots \\ 0.710 & 0.718 & 0.571 & 0.334 & \cdots\cdots & 1 \end{bmatrix}_{8\times8}$ | 8 | $\begin{bmatrix} 1 & 0.941 & 0.488 & 0.536 & \cdots & 0.775 \\ 0.941 & 1 & 0.526 & 0.588 & \cdots & 0.802 \\ 0.488 & 0.526 & 1 & 0.939 & \cdots & 0.627 \\ 0.536 & 0.588 & 0.939 & 1 & \cdots & 0.709 \\ \vdots & \vdots & \vdots & \vdots & \ddots & \vdots \\ 0.775 & 0.802 & 0.627 & 0.709 & \cdots\cdots & 1 \end{bmatrix}_{8\times8}$ |
| 4 | $\begin{bmatrix} 1 & 0.891 & 0.418 & 0.484 & \cdots & 0.647 \\ 0.891 & 1 & 0.465 & 0.532 & \cdots & 0.676 \\ 0.418 & 0.465 & 1 & 0.934 & \cdots & 0.654 \\ 0.484 & 0.532 & 0.934 & 1 & \cdots & 0.702 \\ \vdots & \vdots & \vdots & \vdots & \ddots & \vdots \\ 0.647 & 0.676 & 0.654 & 0.702 & \cdots\cdots & 1 \end{bmatrix}_{8\times8}$ | 9 | $\begin{bmatrix} 1 & 0.922 & 0.292 & 0.300 & \cdots & 0.667 \\ 0.922 & 1 & 0.388 & 0.407 & \cdots & 0.715 \\ 0.292 & 0.388 & 1 & 0.938 & \cdots & 0.607 \\ 0.300 & 0.407 & 0.938 & 1 & \cdots & 0.652 \\ \vdots & \vdots & \vdots & \vdots & \ddots & \vdots \\ 0.667 & 0.715 & 0.607 & 0.652 & \cdots\cdots & 1 \end{bmatrix}_{8\times8}$ |

## Conclusions

This paper demonstrates the ability of the symbolic data representation and the copula-based modelling in improving the performance of image classification, in this case on the MNIST handwritten digit dataset. The pixel intensities are transformed to the empirical cumulative distribution functions (ECDF), and its distribution, a summation of this distribution, called the distribution function of distribution values (DFDV), offering them as an alternative by which local features of an image can be described.

The incorporation of local feature extraction by image scaling and horizontal segmentation, along with cluster analysis to identify prominent distinguishing points, markedly improves classification performance, with an average accuracy of 68.27% and a maximum accuracy of 95.33%. This indicates a significant enhancement compared to previous work, which documented a 62.66% accuracy with a fixed-point selection technique.

**Table 4. Performance comparison between fixed and optimally selected distinguishing points.** The table presents average and maximum classification performance metrics (accuracy, precision, recall, and F1-score) using fixed distinguishing points versus optimally selected points. The proposed method improves average accuracy and recall, indicating better generalization, while maintaining competitive maximum performance levels.

| Partitions | Average (%) | | | | Maximum (%) | | | |
|---|---|---|---|---|---|---|---|---|
| | Accuracy | Precision | Recall | F1-score | Accuracy | Precision | Recall | F1-score |
| Previous (fix distinguishing points) | 62.22 | 65.00 | 62.46 | 63.70 | 96.92 | 95.30 | 95.30 | 95.30 |
| Method propose (optimum distinguishing points) | 68.27 | 62.00 | 66.70 | 63.50 | 95.33 | 90.80 | 95.40 | 93.00 |

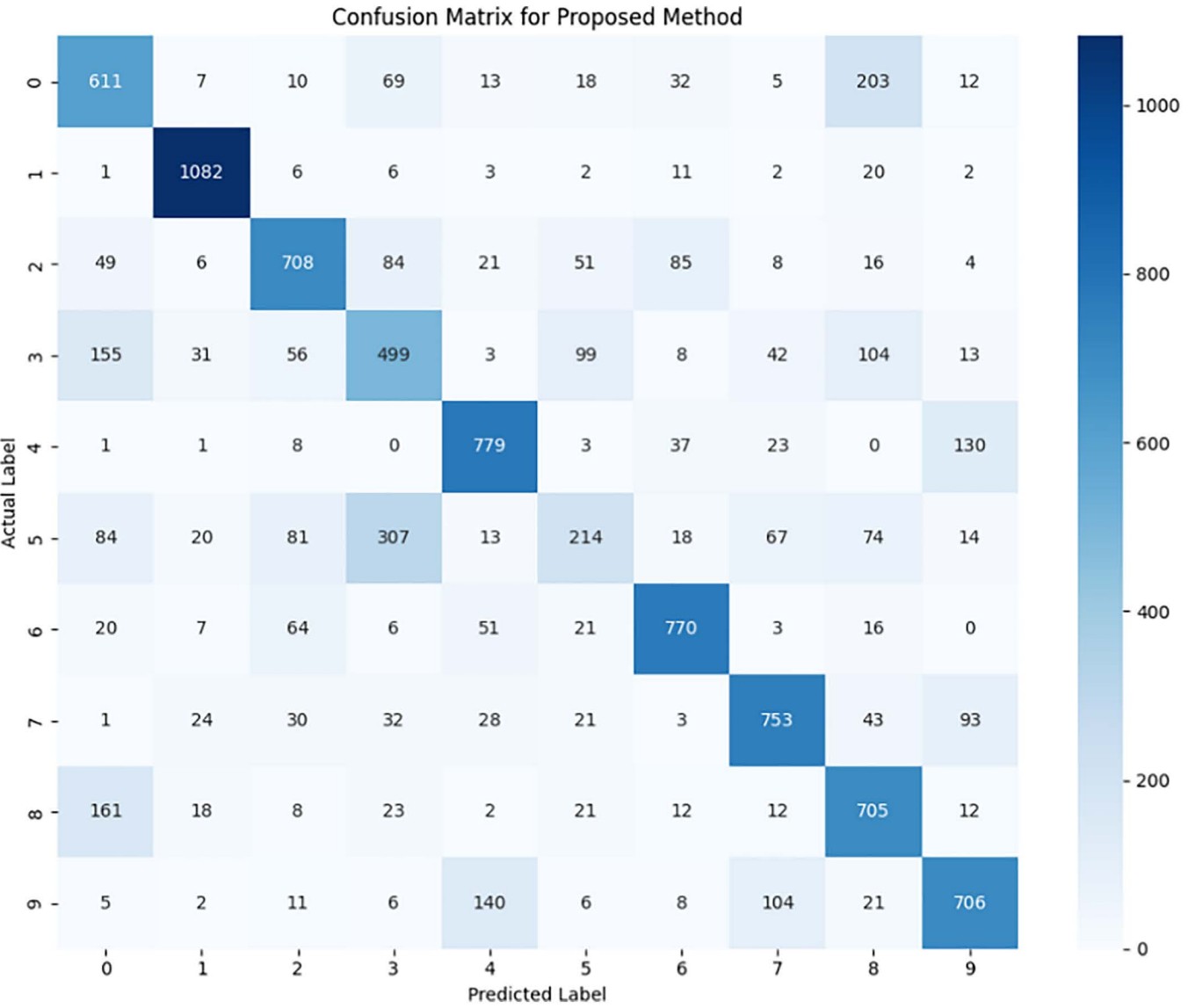

**Fig 8. Confusion matrix for the proposed method using optimally selected distinguishing points.** Cells show the number of predictions for each actual versus predicted digit. Darker colors indicate higher counts, highlighting the most frequent misclassifications.

Moreover, Gaussian copula model displays the connection between local symbolic features, yet KDE begins adaptability in dealing with non-parametric marginal distributions. Although this approach relies on existing segmentation and clustering techniques, it presents a strong potential in the scenarios where interpretability, data organisation and class uniqueness are most important. The results emphasize the possibility of the combination of symbolic data modelling and copula-based framework as an additional method to classical machine learning algorithms.

In this study, we demonstrated that integrating symbolic data with Gaussian copula provides an interpretable approach for MNIST image classification. While the accuracy is slightly lower than CNN and LeNet, our method avoids the "black-box" issue, allowing better understanding of feature interactions through distinguishing points. This approach can be applied in settings where interpretability is critical. Future improvements could focus on optimizing the selection of distinguishing points, combining copula-based interpretability with deep learning accuracy, and extending the method to larger or more complex datasets.

## Acknowledgments

The authors wish to convey their profound appreciation to the anonymous reviewers for their insightful remarks and constructive recommendations, which have markedly enhanced the quality and clarity of this paper. This research was funded by Universitas Padjadjaran, Indonesia, through the Internal Matching Funds Research Grant (IMF) for the project "Model Classification of Rice Plant Diseases Based on Deep Learning and Gaussian Copula to Support Sustainable Precision Agriculture," under contract number 4356/UN6.D/PT.00/2025.

## Author contributions

**Conceptualization:** Sri Winarni, Sapto Wahyu Indratno.

**Data curation:** Sri Winarni, Mohd Murtadha Mohamad, Apri Junaidi, Anindya Apriliyanti Pravitasari.

**Formal analysis:** Sri Winarni, Sapto Wahyu Indratno.

**Funding acquisition:** Sri Winarni, Mohd Shahizan Othman, Irlandia Ginanjar.

**Investigation:** Sri Winarni, Sapto Wahyu Indratno, Mohd Shahizan Othman.

**Methodology:** Sri Winarni, Sapto Wahyu Indratno, Mohd Shahizan Othman, Siti Zaiton Mohd Hashim, Anindya Apriliyanti Pravitasari.

**Project administration:** Mohd Shahizan Othman, Irlandia Ginanjar.

**Resources:** Mohd Shahizan Othman, Siti Zaiton Mohd Hashim, Mohd Murtadha Mohamad, Ebenezer Bonyah.

**Software:** Sri Winarni, Siti Zaiton Mohd Hashim, Apri Junaidi, Triyani Hendrawati.

**Supervision:** Mohd Shahizan Othman.

**Validation:** Mohd Murtadha Mohamad.

**Visualization:** Apri Junaidi, Ebenezer Bonyah, Anindya Apriliyanti Pravitasari, Triyani Hendrawati.

**Writing – original draft:** Sri Winarni, Sapto Wahyu Indratno.

**Writing – review & editing:** Ebenezer Bonyah, Triyani Hendrawati, Irlandia Ginanjar.

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
