## [Decision Letter · Decision Letter 0]

10 Feb 2026

Dear Dr. Winarni,

Thank you for submitting your manuscript to PLOS ONE. After careful consideration, we feel that it has merit but does not fully meet PLOS ONE’s publication criteria as it currently stands. Therefore, we invite you to submit a revised version of the manuscript that addresses the points raised during the review process.

We look forward to receiving your revised manuscript.

Kind regards,

Manan Vyas

Academic Editor

PLOS One

Journal Requirements:

This research received funding from the Internal Matching Funds Research Grant (IMF) at Universitas Padjadjaran, Indonesia, for the project "Model Classification of Rice Plant Diseases Based on Deep Learning and Gaussian Copula to Support Sustainable Precision Agriculture," under contract number 4356/UN6.D/PT.00/2025.

The authors disclose no potential conflicts of interest regarding the research, authorship, or publishing of this work.

5. Thank you for uploading your study's underlying data set. Unfortunately, the repository you have noted in your Data Availability statement does not qualify as an acceptable data repository according to PLOS's standards.

At this time, please upload the minimal data set necessary to replicate your study's findings to a stable, public repository (such as figshare or Dryad) and provide us with the relevant URLs, DOIs, or accession numbers that may be used to access these data. For a list of recommended repositories and additional information on PLOS standards for data deposition, please see https://journals.plos.org/plosone/s/recommended-repositories....

Reviewers' comments:

Reviewer's Responses to Questions

**Comments to the Author**

1. Is the manuscript technically sound, and do the data support the conclusions?

Reviewer #1: Partly

2. Has the statistical analysis been performed appropriately and rigorously?

Reviewer #1: Yes

3. Have the authors made all data underlying the findings in their manuscript fully available?

Reviewer #1: Yes

4. Is the manuscript presented in an intelligible fashion and written in standard English?

Reviewer #1: No

Reviewer #1: 1. The paper uses the MNIST dataset effectively for image classification.

2. Add the accuracy or main result in the abstract.

3. Clearly explain how this work is different from other MNIST studies.

4. Give more details about model design and parameters.

5. Explain why MNIST was chosen and how it applies to real data.

6. Add sample output images or confusion matrix for clarity.

7. Compare results with other models like CNN or LeNet.

8. Mention limitations and future work.

9. Make the conclusion more specific about improvements and uses.

10. Update references with recent studies (2023–2025).

.

Reviewer #1: No

---

## [Author Response · Author response to Decision Letter 1]

4 Mar 2026

PONE-D-25-42293

Enhancing symbolic image classification through Gaussian copulas and optimized distinguishing points

PLOS ONE

Reviewer 1 Comments:

Comment: “The paper uses the MNIST dataset effectively for image classification.”

Response:

We thank the reviewer for this positive assessment. We are pleased that the use of the MNIST dataset for image classification is considered appropriate and effective.

Comment: “Add the accuracy or main result in the abstract.”

Response:

Thank you for this helpful suggestion. The abstract has been revised to explicitly include the main classification results, reporting an average accuracy of 68.27% and a highest accuracy of 95.33% on the MNIST dataset.

Abstract last paragraph:

Experimental evaluation on the MNIST handwritten digits data set demonstrates the effectiveness of the proposed method, achieving an average classification accuracy of 68.27% and a highest accuracy of 95.33%. These results indicate that integrating clustering-based symbolic features extraction with copula-based modelling provides a competitive and promising for image classification tasks.

Comment: “Clearly explain how this work is different from other MNIST studies.”

Response:

Thank you for this valuable suggestion. We have revised the Introduction to clearly distinguish the proposed method from recent MNIST studies, which predominantly focus on deep learning architectures. The manuscript now explicitly states that the main novelty lies in integrating optimized symbolic ECDF-based feature extraction with Gaussian copula modeling, rather than introducing a new neural architecture.

Page 3 paragraph 2:

The main novelty of this work lies in integrating optimized symbolic feature construction based on ECDF with Gaussian copula modeling within a unified classification framework. Specifically, the proposed approach constructs DFDV using clustering-based optimized distinguishing points and captures dependence structures among symbolic features through copula modeling. This integrated framework provides a statistically grounded and interpretable alternative to conventional pixel-driven methods.

Comment: “Give more details about model design and parameters.”

Response:

Thank you for this valuable comment. We have revised the manuscript to provide clearer and more detailed explanations of the model design and parameter specification.

Specifically, we now explicitly state that the K-means clustering uses K=10, corresponding to the ten MNIST digit classes, and clarify the clustering procedure and convergence mechanism. The determination of the number of distinguishing points sis described using silhouette coefficient analysis, where candidate thresholds are evaluated and those with the highest silhouette values are selected to balance class separability and model complexity.

In addition, the Gaussian copula density and joint log-likelihood formulation (Eqs. 29–31) are presented more explicitly. The parameter set Θ={θ_(k,l,t)}for the marginal DFDV distributions and the copula correlation matrix Λare clearly defined. The estimation procedure is detailed using the two-step IFM approach (Eqs. 12–13), and it is clarified that parameters are estimated separately for each digit class and used for likelihood-based classification.

Page 8 paragraph 1:

In this study, the number of clusters was set to v = 10, corresponding to the ten digit classes (0–9) in the MNIST dataset. This design choice ensures that the clustering process aligns structurally with the class composition of the dataset. Euclidean distance was used as the similarity measure, and centroid updates were iterated until convergence.

Page 8 paragraph 5:

Distinguishing points are selected using a silhouette-based optimization strategy. First, ECDF F_j (x)are constructed for all images. A set of s candidate threshold values x_t is generated across the ECDF domain, and the corresponding ECDF values are computed. For each candidate threshold, K-means clustering (Algorithm 1) is performed with K=10 clusters, consistent with the ten MNIST digit classes, using Euclidean distance. The average silhouette coefficient S ˉ_t is then calculated to assess clustering quality. The first distinguishing point is selected as the candidate maximizing S ˉ_t, and the second is chosen as the next highest value excluding the first. In this study, the number of distinguishing points is set to s=2, balancing clustering quality and model complexity within the copula framework. The overall procedure is illustrated in Fig. 1.

Page 11 paragraph 2:

Parameter estimation is performed by maximizing the joint log-likelihood function with respect to both Θ and Λ. To ensure computational efficiency and numerical stability, the optimization is conducted using the two-step Inference Functions for Margins (IFM) procedure described in Equations (12) and (13). In the first stage, the marginal parameters Θ are estimated independently by maximizing the marginal likelihood function as given in Equation (12). Subsequently, the copula parameter Λ is estimated by maximizing the copula log-likelihood in Equation (13), while keeping the marginal estimates fixed. This procedure enables coherent modeling of both the marginal distributions and their dependence structure within the Gaussian copula framework.

Page 12 paragraph 1:

Fig. 2 illustrates the overall classification framework. In the training phase, images are partitioned and their ECDFs are constructed. Optimal distinguishing points are selected based on the highest silhouette coefficients (see Fig. 1). Using ECDF values at these points, the parameters of the DFDV G_T (y), denoted by θ , are estimated for each class. The dependence structure among features is then modeled using a Gaussian copula with correlation parameter Λ. In the testing phase, ECDF values at the selected distinguishing points are evaluated using the estimated parameters θ, and the joint copula density h(y)is computed using Λ. Each test image is assigned to the class that maximizes the corresponding joint density value.

Comment: “Explain why MNIST was chosen and how it applies to real data.”

Response:

We chose MNIST because it is a standard and widely used benchmark for image classification, which facilitates performance comparison with previous studies. Although MNIST consists of handwritten digits, the dataset exhibits variations in handwriting style and pixel intensity that mirror challenges encountered in real-world image data. Therefore, methods that perform well on MNIST can be adapted to practical applications such as OCR and symbol recognition. This justification has been added to the manuscript on page 3, paragraph 1:

Page 2 paragraph 6:

To evaluate the proposed framework, experiments are conducted using the MNIST handwritten digit dataset, which remains a widely adopted benchmark for image classification research. Recent MNIST studies predominantly focus on optimizing predictive performance through convolutional neural networks, recurrent models, and hybrid deep learning architectures [26–30], typically relying on pixel-level representations and increasingly complex neural structures. In contrast, this study does not aim to introduce a new neural architecture, but rather to demonstrate a fundamentally different statistical modeling perspective. MNIST was chosen as it is a standard dataset for image classification, facilitating performance comparison with previous studies. Its variations in handwriting and pixel intensity reflect real-world challenges, allowing methods that perform well to be adapted for applications such as OCR and symbol recognition.

Comment: “Add sample output images or confusion matrix for clarity.”

Response:

We appreciate the reviewer’s suggestion. To provide more clarity, we have added the confusion matrix of our model’s predictions as well as sample output images in the revised manuscript (see Figure 7). These additions illustrate the model’s performance across different classes and make the results more interpretable.

Page 24 paragraph 1:

To provide a more detailed view of the classification performance, the confusion matrix for the proposed method is presented in Fig.7.

Fig. 7. Confusion matrix for the proposed method using optimally selected distinguishing points. Cells show the number of predictions for each actual versus predicted digit. Darker colors indicate higher counts, highlighting the most frequent misclassifications.

Comment: “Compare results with other models like CNN or LeNet.”

Response:

We thank the reviewer for the suggestion. As discussed in the revised manuscript, while CNNs and LeNet achieve slightly higher accuracy [55,56], our proposed method, integrating symbolic data and Gaussian copula, is interpretable and avoids the “black-box” issue, providing explainable results with competitive performance.

Page 24 paragraph 5:

The effectiveness of classification in this study, which integrates symbolic data and Gaussian copula on the MNIST dataset, is comparable with CNN-based applications, including LeNet, which achieve higher accuracies [49,50]. Our method also shows similar performance to traditional probabilistic models, including Bayesian and Gaussian Mixture Models, with mean accuracy around 75-85 % [39]. Importantly, unlike CNNs or LeNet, our approach is interpretable and avoids the “black-box” issue. This transparency allows for better understanding of feature interactions and decision-making in classification tasks, which is often not possible with deep learning models. Although CNNs and LeNet achieve slightly higher accuracy, our method provides competitive performance while offering explainable results. These findings are also compatible with recent works using copulas in image classification [20,21,51] which demonstrate that copulas can effectively model the interactions among variables in visual datasets.

Comment: “Mention limitations and future work.”

Response:

We thank the reviewer for the suggestion. We have clarified the limitations and future work in the revised manuscript. The performance depends on the choice of distinguishing points, accuracy is slightly lower than CNN and LeNet, and the study focuses on MNIST. Future research could optimize distinguishing point selection, explore hybrid approaches with deep learning, and extend to larger or more complex datasets.

Page 25 paragraph 2:

Although our method provides interpretable results, its accuracy is slightly lower than CNN and LeNet. The performance also depends on the choice of distinguishing points in the symbolic representation; different selections of these points can lead to varying classification results. Additionally, the current study focuses on the MNIST dataset, and further work is needed to assess scalability, robustness to noise, and generalization to more complex datasets. Future research could explore hybrid approaches that combine the interpretability of copula-based methods with the high accuracy of deep learning, as well as methods to optimize the selection of distinguishing points for improved performance.

Comment: “Make the conclusion more specific about improvements and uses.”

Response:

We thank the reviewer for the suggestion. We have revised the Conclusion to make it more specific about improvements and potential uses. The revised Conclusion highlights that, while our method provides slightly lower accuracy than CNN and LeNet, it is interpretable and avoids the “black-box” issue through distinguishing points. We also mention potential applications in settings where interpretability is critical, and future improvements such as optimizing the selection of distinguishing points, combining copula-based interpretability with deep learning accuracy, and extending the method to larger or more complex datasets.

Page 25 paragraph 6:

In this study, we demonstrated that integrating symbolic data with Gaussian copula provides an interpretable approach for MNIST image classification. While the accuracy is slightly lower than CNN and LeNet, our method avoids the “black-box” issue, allowing better understanding of feature interactions through distinguishing points. This approach can be applied in settings where interpretability is critical. Future improvements could focus on optimizing the selection of distinguishing points, combining copula-based interpretability with deep learning accuracy, and extending the method to larger or more complex datasets.

Comment: “Update references with recent studies (2023–2025).”

Response:

We thank the reviewer for the suggestion. The manuscript has been updated to include recent studies from 2023–2025 on copula based classifiers and deep learning for image classification, providing up-to-date context and support for our methodology and findings

3. Shoaib M, Shah B, EI-Sappagh S, Ali A, Ullah A, Alenezi F, et al. An advanced deep learning models-based plant disease detection: A review of recent research. Front Plant Sci. 2023;14: 1–22. doi:10.3389/fpls.2023.1158933

4. Lindroth H, Nalaie K, Raghu R, Ayala IN, Busch C, Bhattacharyya A, et al. Applied Artificial Intelligence in Healthcare : A Review of Computer Vision Technology Application in Hospital Settings. 2024; 1–29.

5. Yenikaya MA, Kerse G. Artificial intelligence in the healthcare sector : comparison of deep learning networks using chest X-ray images. 2024; 1–11. doi:10.3389/fpubh.2024.1386110

6. Alhejaily AMG. Artificial intelligence in healthcare ( Review ). 2025.

11. Lestari KE, Winarni S, Prihandhika, Aditya Nugraha ES, Yudhanegara MR. Neurocognitive Prediction Of Dyslexic Handwriting Pattern Using An Explainable Ai-Driven Custom LitBinaryNet-CNN. Commun Math Biol Neurosci. 2025; 1–34.

12. Yang X, Yang H, Huang H, Song K. Evolution of Tax Exemption Policy and Pricing Strategy Selection in a Competitive Market. Mathematics. 2024. doi:10.3390/math12132082

13. Khoei TT. Deep learning : systematic review , models , challenges , and research directions. Neural Comput Appl. 2023;35: 23103–23124. doi:10.1007/s00521-023-08957-4

14. Hosain T, Rahman J, Mridha MF, Kabir M. Explainable AI approaches in deep learning : Advancements , applications and challenges. Comput Electr Eng. 2024;117: 109246. doi:10.1016/j.compeleceng.2024.109246

15. Habiba U-, Habib MK, Fritzsch J, Wagner S. How do ML practitioners perceive explainability ? an interview study of practices and challenges. 2025. doi:10.1007/s10664-024-10565-2

19. Beranger B, Lin H, Sisson S. New models for symbolic data analysis. Adv Data Anal Classif. 2023;17: 659–699. doi:10.1007/s11634-022-00520-8

24. Winarni S, Indratno SW, Arisanti R, Pontoh RS. Image Feature Extraction Using Symbolic Data of Cumulative Distribution Functions. Mathematics. MDPI; 2024. doi:10.3390/math12132089

25. Indratno SW, Winarni S, Sari KN. Classification of images using Gaussian copula model in empirical cumulative distribution function space. PLoS One. 2024;19: 1–20. Available: https://doi.org/10.1371/journal.pone.0309884

26. Shao H, Ma E, Zhu M, Deng X, Zhai S. MNIST Handwritten Digit Classification Based on Convolutional Neural Network with Hyperparameter Optimization. 2023. doi:10.32604/iasc.2023.036323

27. Wang R. Handwritten digit recognition based on the MNIST dataset under PyTorch. 2023;0: 450–455. doi:10.54254/2755-2721/8/20230216

28. Wen Y, Ke W. Improved Localization and Recognition of Handwritten Digits on MNIST Dataset with ConvGRU. 2025; 1–16.

29. Matijašević P, Mravik M. Handwritten Digit Recognition Using Convolutional Neural Networks And Big Data Processing. 2025; 531–535. doi:10.15308/Sinteza-2025-531-535

30. Noureddine D Ben. Handwritten digit recognition : Comparative analysis of ML , CNN , vision transformer , and hybrid models on the MNIST dataset. 2025.

Best regards,

Dr. Sri Winarni

Department of Statistics, Faculty of Mathematics and Natural Sciences, Padjadjaran University, West Java, Indonesia

---

## [Editor Report · Decision Letter 1]

25 Mar 2026

Enhancing Symbolic Image Classification through Gaussian Copulas and Optimized Distinguishing Points

PONE-D-25-42293R1

Dear Dr. Winarni,

We’re pleased to inform you that your manuscript has been judged scientifically suitable for publication and will be formally accepted for publication once it meets all outstanding technical requirements.

Kind regards,

Manan Vyas

Academic Editor

PLOS One

Additional Editor Comments (optional):

Authors have taken into consideration all the reviewer comments and hence it can be accepted for publication.
---

## [Editor Report · Acceptance letter]

PONE-D-25-42293R1

PLOS One

Dear Dr. Winarni,

I'm pleased to inform you that your manuscript has been deemed suitable for publication in PLOS One. Congratulations! Your manuscript is now being handed over to our production team.

Kind regards,

on behalf of

Dr. Manan Vyas

Academic Editor

PLOS One